# Cell Painting PLUS: expanding the multiplexing capacity of Cell Painting-based phenotypic profiling using iterative staining-elution cycles

Elena von Coburg [1,2], Marlene Wedler [1,3], Jose M. Muino [1,4], Christopher Wolff [5], Nils Körber [6], Sebastian Dunst [1,7] ✉ & Shu Liu [1,7] ✉

Phenotypic changes in the morphology and internal organization of cells can indicate perturbations in cell functions. Therefore, imaging-based high-throughput phenotypic profiling (HTPP) applications such as Cell Painting (CP) play an important role in basic and translational research, drug discovery, and regulatory toxicology. Here we present the Cell Painting PLUS (CPP) assay, an efficient, robust and broadly applicable approach that further expands the versatility of available HTPP methods and offers additional options for addressing mode-of-action specific research questions. An iterative staining-elution cycle allows multiplexing of at least seven fluorescent dyes that label nine different subcellular compartments and organelles including the plasma membrane, actin cytoskeleton, cytoplasmic RNA, nucleoli, lysosomes, nuclear DNA, endoplasmic reticulum, mitochondria, and Golgi apparatus. In this way, CPP significantly expands the flexibility, customizability, and multiplexing capacity of the original CP method and, importantly, also improves the organelle-specificity and diversity of the phenotypic profiles due to the separate imaging and analysis of single dyes in individual channels.

Phenotypic screening using imaging-based high-throughput phenotypic profiling (HTPP) facilitates the efficient extraction and analysis of a broad range of cellular phenotypes induced by compounds or genetic perturbations in basic and translational research, drug discovery, and regulatory toxicology[1–3]. The HTPP concept is essentially based on the assumptions that changes in morphology and organization of sub-/cellular structures can indicate perturbations in cell functions and that compounds with similar mode-of-action (MoA) deliver similar phenotypic profiles.

Cell Painting (CP) is a widely utilized HTPP method for multiplexed staining and analysis of subcellular compartments and organelles including nuclear DNA, cytoplasmic RNA, nucleoli, endoplasmic reticulum (ER), actin cytoskeleton (Actin), Golgi apparatus (Golgi), plasma membrane, and mitochondria (Mito)[4–7]. The CP method involves staining of cells with fluorescent dyes and subsequent high-content imaging to generate multidimensional image datasets. Those images are fed into automated image and data analysis routines that extract quantitative information about sub-/cellular morphological

[1]German Centre for the Protection of Laboratory Animals (Bf3R), German Federal Institute for Risk Assessment (BfR), Berlin, Germany. [2]Department of Food Chemistry, University of Potsdam, Potsdam, Germany. [3]Institute of Biology, Free University of Berlin, Berlin, Germany. [4]Institute of Clinical Pharmacology and Toxicology, Charité – Universitätsmedizin Berlin, corporate member of Freie Universität Berlin, Humboldt-Universität zu Berlin, and Berlin Institute of Health, Berlin, Germany. [5]Screening Unit, Leibniz-Forschungsinstitut für Molekulare Pharmakologie (FMP), Berlin, Germany. [6]Centre for Artificial Intelligence in Public Health Research, Robert Koch Institute, Berlin, Germany. [7]These authors contributed equally: Sebastian Dunst, Shu Liu. ✉e-mail: sebastian.dunst@bfr.bund.de; shu.liu@bfr.bund.de

features for generation of morphological profiles barcoding compound activities or genetic perturbations[8]. Thus, unlike many other targeted bioassays that measure specific, expected phenotypic responses of cells to perturbations in cellular functions[9,10], CP enables the generation of broad phenotypic profiles at single-cell resolution in an untargeted manner. This way, CP supports the identification of compounds or genetic perturbations with similar MoA in a predefined cellular context as well as the identification of distinct cell type-specific activities.

Leveraging its high-throughput screening (HTS) capacity, CP has already been used in the context of hazard assessment of industrial chemicals to generate bioactivity profiles for more than 1,000 industrial chemicals in human U2OS osteosarcoma cells, which have been taken up into the public U.S. EPA CompTox Chemicals Dashboard[11,12]. Moreover, the *Joint Undertaking for Morphological Profiling* (JUMP)-Cell Painting Consortium[13] generated the currently largest dataset of images and phenotypic profiles for more than 135,000 (pharmaceutical) compounds and genetic perturbations including over-expression and knockout of genes using CRISPR-Cas9 in U2OS cells[14,15], which are publicly available through the *Cell Painting Gallery*. This work is now being followed up by the OASIS Consortium[16], which uses hepatotoxicity as a use-case for combining and benchmarking phenomics, transcriptomics, and proteomics data obtained from U2OS or primary human HepaRG liver cells against existing rat and human in vivo data to increase confidence in the physiological relevance of the cellular responses measured by CP.

The standardization of CP needed for those large-scale HTS projects is, however, also accompanied by a strong tendency to examine only a scarce number of different cell types under sub-confluent conditions, which arguably represents advantageous conditions for robust spatial imaging. However, this also limits the physiological relevance and mechanistic diversity of available phenotypic profiling datasets, with biologically more diverse cell culture conditions being mainly applied only in more specialized, smaller-scale screening studies[17], thereby leaving cell type-specific (e.g., nuclear hormone receptor pathway-mediated) responses rather unattended. Moreover, CP was designed for cost- and time-efficient HTPP using a fixed set of dyes for selected subcellular compartments and organelles acquired in typically four to five imaging channels with the commonly used high-content imaging systems[4,5,18,19]. To maximize HTPP capacity while maintaining a very high information density, signals from two CP dyes are often intentionally merged in the same imaging channel (i.e., RNA + ER and/or Actin+Golgi)[4,5,11,12,14,17–27], accepting the trade-off that this optimization may compromise the organelle-specificity of the phenotypic profiles.

To complement the versatile HTPP methods with additional options for addressing more MoA-specific research questions, we therefore developed the Cell Painting PLUS (CPP) assay. CPP significantly expands the flexibility, customizability, and multiplexing capacity of the original CP method, and, importantly, also improves the organelle-specificity and diversity of the phenotypic profiles. In CPP, an optimized elution buffer enables the iterative staining, elution, and re-staining of single cells, thereby providing more flexibility in selecting and combing various fluorescent dyes for diverse subcellular compartments and organelles to customize the method according to the specific research questions. CPP also enables fully sequential imaging of each dye in a separate channel, thereby achieving spectral signal separation and generation of more specific phenotypic profiles to gain more precise insights into cellular processes and functional perturbations. One additional advantage of CPP over CP is, that it can be adapted to individual needs (e.g., using set of dyes or even antibodies specific to a certain research question). Thus, CPP can indeed be used as a very specific, customizable screening method that expands the repertoire of the already existing, valuable HTPP methods such as CP.

## Results

### CPP expands the multiplexing capacity of the CP method

Using the hormone-responsive MCF-7/vBOS (abbreviated as MCF-7) breast cancer cell line, we further developed the original Cell Painting (CP) method[4,5] into the Cell Painting PLUS (CPP) assay. When conducting CP in this comparative study, we followed a typical imaging set-up using four laser lines and captured dyes with largely overlapping spectral ranges, i.e., RNA/ER and Actin/Golgi (AGP), in the same imaging channel (Fig. 1A). In contrast, in the CPP assay, all dyes were captured in separate imaging channels, providing more specific information for these organelles (Fig. 1B). Notably, CPP also included the additional staining of lysosomes.

With regard to the selection and evaluation of suitable dyes for CPP, we systematically investigated spectral crosstalk (emission bleed-through and cross-excitation) and the signal stability of dyes over time. Emission bleed-through was observed for the RNA dye and, to a weaker extent, for the DNA dye when excited with the 488 nm or 405 nm imaging lasers, respectively (Supplementary Fig. 1A). Therefore, each dye was sequentially imaged in the CPP assay to avoid effects of emission-bleed-through on staining specificity. The RNA dye also showed some cross-excitation when excited with the Mito channel laser at 561 nm (Supplementary Fig. 1A). This property of the RNA dye leading to weak emission in the Mito channel, could not be fully mitigated by adapting imaging routines but was considered to be minor due to the high signal-to-noise ratio of the specific Mito signal. However, as a precautionary measure, the Mito signal was only analyzed in the staining second cycle as summarized below.

All CPP dyes were well detectable over the course of 4 weeks after staining, but the staining intensities of all dyes remained sufficiently stable only until day 1 (deviation of less than ± 10% compared to day 0) (Supplementary Fig. 1A). Prominent differences in the relative signal intensities over time were observed for the Lyso dye (decreasing) and the ER dye (increasing) already from day 2 after staining. This may be related to pH-dependent changes in the fluorescence properties of the dye itself, or due to changes in the binding of the organelle-targeting moiety. The LysoTracker™ dye accumulates in lysosomes due to their acidic pH (~4.5–5.0) and is therefore being applied to live cells. In turn, concanavalin A is used in fixed cells and binds to specific carbohydrate structures, such as mannose and glucose residues, that are enriched at the ER. After cell fixation by cross-linking proteins and other cellular components with paraformaldehyde (PFA), the cellular morphology is preserved, and the ability of fluorescent dyes to diffuse within the cell should be significantly limited, but is not fully inhibited. Under the staining conditions used in this study, the Lyso and ER dyes may take longer (up to day 2) to fully reach equilibrium. Whereas the signal intensity of the ER dye is maintained at this plateau, the intensity of Lyso may have quickly dropped again to lower levels after day 2. This emphasizes that thoroughly characterizing the fluorescence properties of the dyes used in the specific experimental setting is crucial for interpreting the results, especially when using dyes in different cellular compartments with distinct pH levels, such as in the highly acidic environment of lysosomes. For that reason, imaging was conducted in the CPP assay within 24 h after staining to ensure robustness of phenotypic profiling data. The concentrations of the CPP dyes and corresponding exposure times were chosen to achieve a balanced compromise between dye cost and total imaging time, while maintaining an optimal signal intensity range (Supplementary Fig. 1B, Supplementary Data 1). The dye concentrations used in CP and in the original[4] or recently updated[5] CP method were similar indicating comparable screening costs in CPP and CP per single dye used (Supplementary Data 1). Thus, the additional reagent costs of the CPP assay are mainly due to the inclusion of the Lyso dye (Supplementary Data 1), but may decrease if alternative Lyso dyes compatible with fixed cell staining become available in the future.

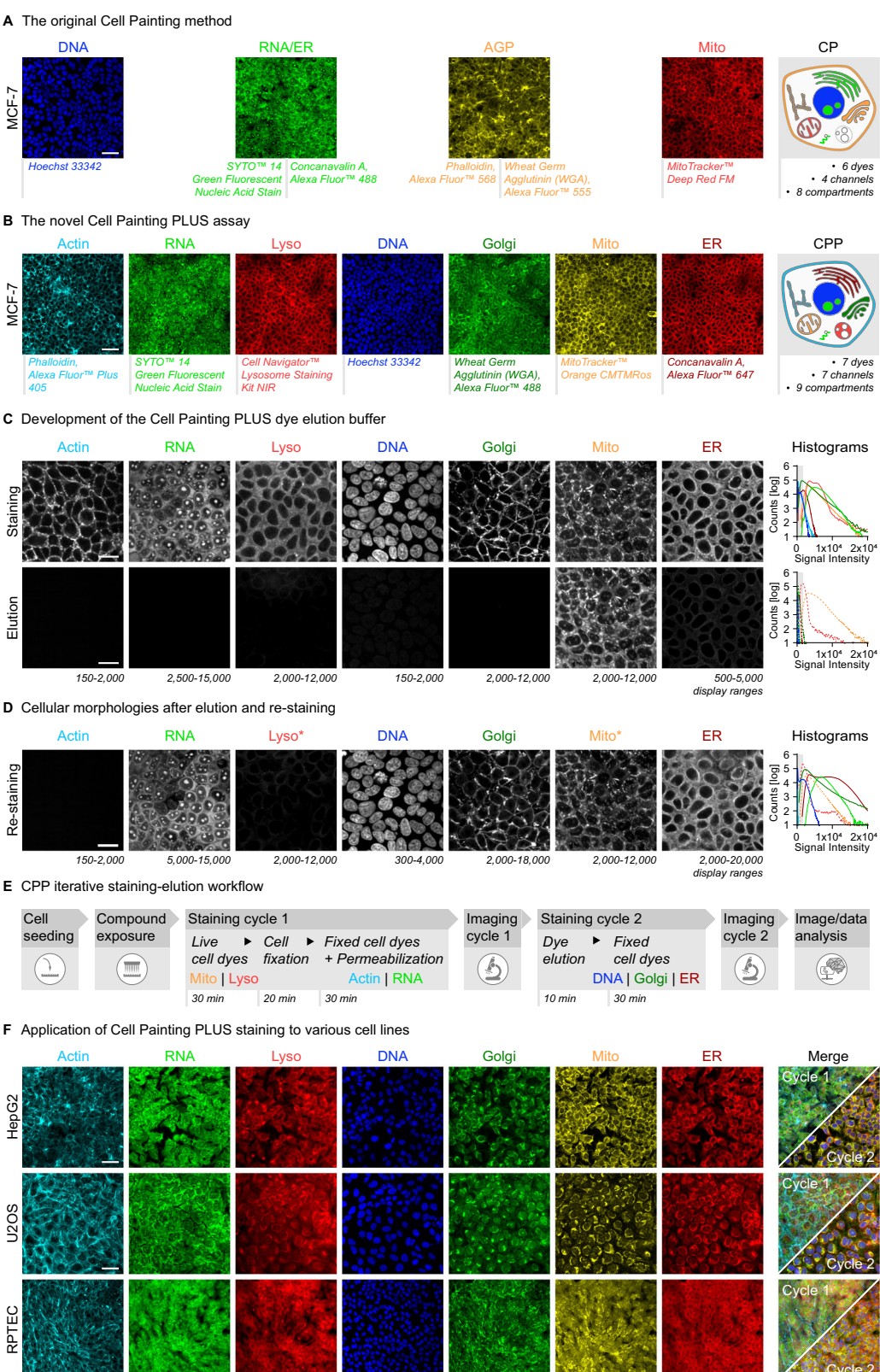

**Fig. 1 | CPP assay enables multiplexed imaging of cellular structures based on iterative staining-elution cycles. A**, **B** Representative images and schematic illustration comparing the original Cell Painting (CP) method to the Cell Painting PLUS (CPP) assay conducted in MCF-7 cells. $N_{Biol}$ = 4. Scale bars = 50 μm. **C** Representative images and corresponding histograms showing single dye stainings before and after elution using the CPP dye elution buffer in MCF-7 cells. Signal intensity ranges of displayed images are indicated enabling comparison between channels and to data shown in Fig. 2C. $N_{Biol}$ = 1. Scale bars = 20 μm. **D** Representative

images and corresponding histograms showing cellular morphologies after elution and re-staining of MCF-7 cells shown in (**C**) using the same dyes (* except for live-cell dyes). Signal intensity ranges of displayed images are indicated enabling comparison between channels and to data shown in (**C**). $N_{Biol}$ = 1. Scale bars = 20 μm. **E** Schematic workflow outlining the CPP staining and elution procedure. **F** Representative images showcasing the application of CPP staining to U2OS, HepG2, and RPTEC cells. $N_{Biol}$ = 4. Scale bars = 50 μm.

## CPP enables iterative staining-elution cycles using an efficient dye elution buffer

To allow iterative staining of cells, we developed a dye elution buffer that efficiently removed the staining signals but preserved sub-cellular compartment and organelle morphologies (Fig. 1C, D). The development and optimization of suitable elution buffers for each dye involved extensive testing of various buffer components and parameters in combination including different pH, reducing agents, chaotropic agents (ionic strength), temperatures, and elution times. The optimal elution buffer compositions for each dye are summarized in Supplementary Data 1 to guide other laboratories in implementing and customizing their own CPP assay depending on their specific needs. Notably, the CPP elution buffer (0.5 M L-Glycine, 1% SDS, pH 2.5), which was used in this study for all phenotypic profiling screens, was designed to efficiently remove the signals of all dyes except for the Mito dye (Fig. 1C). This was intended to be able to use the Mito channel in the image analysis workflow as a reference channel for combination (registration) of individual image stacks from multiple staining cycles into a single multi-channel image stack (Fig. 2B). Notably, the final CPP elution buffer demonstrated superior efficacy for dye elution in comparison to other published buffers[28–30] that were used for antibody elution (Supplementary Fig. 1C). Thus, the CPP elution buffer provided not only an efficient, but also a time- and cost-effective elution procedure in the CPP protocol (Supplementary Data 1).

To investigate whether the CPP elution step may influence subcellular compartment and organelle morphologies or interfere with dye binding, we re-stained the fixed cells of staining cycle 1 after elution with exactly the same dye in a staining cycle 2, except for the live-cell Lyso and Mito dyes. All dyes but not the Actin dye could be re-stained and the respective morphologies of subcellular compartments and organelles were preserved (Fig. 1D). The Mito dye was not eluted and showed very comparable signal intensities in staining cycles 1 and 2. Overall signal intensities of some re-stained dyes slightly differed between the two staining cycles. Regarding the absence of Actin dye signals after re-staining, testing of other Phalloidin dyes and an anti-Actin antibody revealed that the CPP elution step might generally inhibit the binding of Phalloidin-based dyes to actin fibers rather than disrupting the actin cytoskeleton (Supplementary Fig. 1D). Thus, in the final CPP staining-elution workflow used in this study (Fig. 1E), the two live-cell Lyso and Mito dyes were included in staining cycle 1 along with the Actin dye (to avoid re-staining issues after elution) and the RNA dye (slightly higher elution efficiency than the Golgi dye). Accordingly, the DNA, Golgi, and ER dyes were assigned to staining cycle 2 and analyzed together with the remaining Mito signal (to avoid cross-excitation of the RNA dye) from cycle 1.

To demonstrate transferability of this staining-elution approach and the broad applicability of the CPP assay, we tested three additional human cell lines that are widely used as surrogate models for relevant target organs including the U2OS (bone) and HepG2 (liver) cancer cell lines as well as post-mitotic, differentiated RPTEC-TERT1 (abbreviated as RPTEC) primary kidney cells (Fig. 1F). For each cell line, the seeding density was optimized to achieve ~80% confluency for HepG2 and U2OS cells or full confluency for MCF-7 and RPTEC at the time of imaging. For the epithelial cell lines MCF-7 and RPTEC, cells were imaged at full confluency to more closely reflect physiologically-relevant conditions resulting in a smaller size and more compact cytoplasm as compared to HepG2 and U2OS cells, which were less densely seeded and grew more flatly to maintain comparability to publicly available CP data. The adaptation of CPP to different cell lines did not need any further modifications to the staining protocol or the image/data analysis pipelines, indicating the versatile applicability of CPP to diverse cellular and biological contexts.

## Barcoding compound effects using CPP expands the diversity of phenotypic profiles

In order to compare CPP with the original CP method, we established a reference compound plate comprising 13 diverse compounds (drugs, biotoxins, plant alkaloids, and industrial chemicals) that target relevant organelles and cell functions[17,31–44], as well as two compounds (saccharin and sorbitol) serving as negative controls along with the DMSO solvent control (Fig. 2A). Eight concentrations (at half-log dilution) of each reference compound were distributed across the reference compound plate in three blocks of technical replicates and alternating order to evaluate potential plate position effects (Supplementary Data 2).

Using this reference compound plate, we conducted small-scale CPP screens with 48 h exposure time in MCF-7, HepG2, U2OS, and RPTEC cells. For direct comparison of the CPP profiling data with CP, we also conducted the reference compound screen using the CP method in MCF-7 cells to explore its specific added value for phenotypic profiling. The data was analyzed using customized image and data analysis workflows as outlined for CPP in Fig. 2B. The CPP and CP image analysis procedures included the registration and segmentation of images (Supplementary Fig. 2A), followed by the extraction of cell features using the commercial Harmony software (Revvity Inc.) (CPP/Harmony: 894 features, CP/Harmony: 558 features), representing the primary image analysis software that was applied to all cell lines in this study. For comparison, we additionally analyzed CPP images in MCF-7 cells in an analogous way using the open-source Cell Profiler software[45] (CPP/Cell Profiler: 3648 features) as an alternative image analysis method. The extracted cell features were subsequently fed into the data analysis procedure, which included the standardization and visualization of the cell feature data using the open-source KNIME process automation software[46] to generate compound activity profiles (see Fig. 2) and conduct benchmark concentration (BMC) modeling (see Fig. 3). For CPP/Cell Profiler, standardization of cell features was followed by an additional step of feature selection/reduction to align with common practice[47] (CPP/Cell Profiler: 345 selected features).

The compound activity profiles of the CPP reference compound screen revealed distinct concentration-dependent activity patterns for the four cell lines (Fig. 2C, Supplementary Fig. 2B, Supplementary Data 3, Supplementary Data 12–18). Very strong cellular responses spanning multiple channels often correlated with cytotoxic concentration ranges, i.e., compound concentrations that led to a reduction of the relative cell number below 53% compared to the DMSO solvent control (Fig. 2C, Supplementary Data 8–11). This 53% threshold for cytotoxicity was set to ensure that the maximum non-cytotoxic concentrations of the reference compounds were identical between CPP and CP in order to enable direct comparisons of the two methods in MCF-7 cells. The four cell lines showed different cytotoxicities for the reference compounds, with post-mitotic RPTEC cells being the least sensitive. As expected, the activity profiles of the negative control compounds saccharin and sorbitol did not indicate relevant activities across all cell lines in CPP and CP (Fig. 2C, D, Supplementary Fig. 2B). Notably, the activity profiles of fluphenazine, siramesine as well as tetrandrine displayed prominent effects in the Lyso channel at non-cytotoxic concentrations across all cell lines. Further, the weaker activity profile of tetrandrine at the highest tested concentration indicated potential compound solubility issues across all cell lines. The resulting lower exposure levels were also reflected in the corresponding relative cell number plots (Supplementary Data 3–6), consequently leading to the exclusion of this concentration from subsequent analysis. The profile of cytochalasin D was pronounced in the Actin channel and the profile of berberine chloride in the Mito channel, particularly in HepG2, U2OS, and RPTEC cells. Overall, CPP and CP showed largely consistent compound activity profiles for the DNA and Mito channels common to both methods (Supplementary

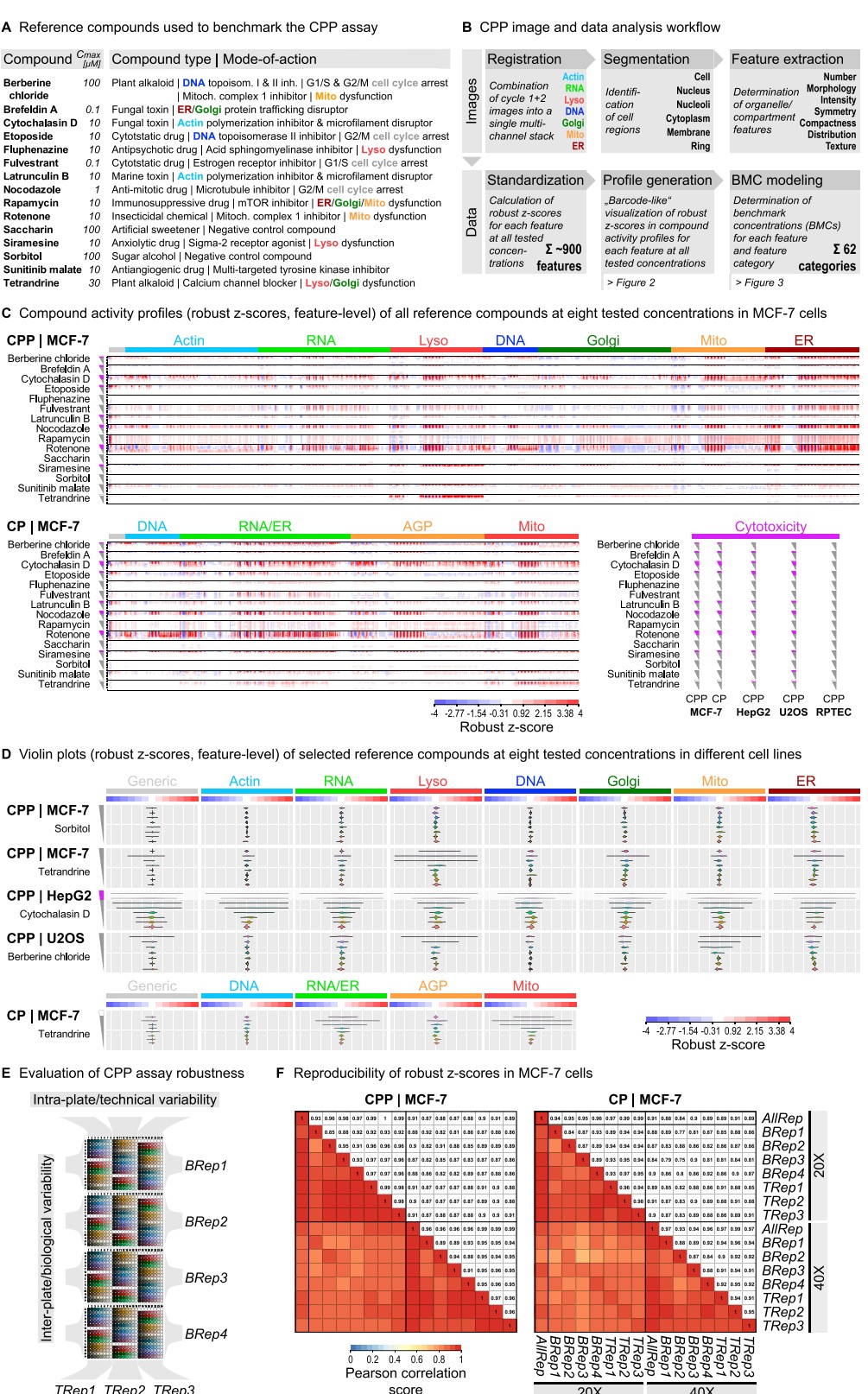

**A** Reference compounds used to benchmark the CPP assay

| Compound | $C_{max}$ [µM] | Compound type \| Mode-of-action |
|---|---|---|
| Berberine chloride | 100 | Plant alkaloid \| **DNA** topoisom. I & II inh. \| G1/S & G2/M **cell cylce** arrest \| Mitoch. complex 1 inhibitor \| **Mito** dysfunction |
| Brefeldin A | 0.1 | Fungal toxin \| **ER**/**Golgi** protein trafficking disruptor |
| Cytochalasin D | 10 | Fungal toxin \| **Actin** polymerization inhibitor & microfilament disruptor |
| Etoposide | 10 | Cytostatic drug \| **DNA** topoisomerase II inhibitor \| G2/M **cell cycle** arrest |
| Fluphenazine | 10 | Antipsychotic drug \| Acid sphingomyelinase inhibitor \| **Lyso** dysfunction |
| Fulvestrant | 0.1 | Cytostatic drug \| Estrogen receptor inhibitor \| G1/S **cell cycle** arrest |
| Latrunculin B | 10 | Marine toxin \| **Actin** polymerization inhibitor & microfilament disruptor |
| Nocodazole | 1 | Anti-mitotic drug \| Microtubule inhibitor \| G2/M **cell cycle** arrest |
| Rapamycin | 10 | Immunosuppressive drug \| mTOR inhibitor \| **ER**/**Golgi**/**Mito** dysfunction |
| Rotenone | 10 | Insecticidal chemical \| Mitoch. complex 1 inhibitor \| **Mito** dysfunction |
| Saccharin | 100 | Artificial sweetener \| Negative control compound |
| Siramesine | 10 | Anxiolytic drug \| Sigma-2 receptor agonist \| **Lyso** dysfunction |
| Sorbitol | 100 | Sugar alcohol \| Negative control compound |
| Sunitinib malate | 10 | Antiangiogenic drug \| Multi-targeted tyrosine kinase inhibitor |
| Tetrandrine | 30 | Plant alkaloid \| Calcium channel blocker \| **Lyso**/**Golgi** dysfunction |

**B** CPP image and data analysis workflow

**C** Compound activity profiles (robust z-scores, feature-level) of all reference compounds at eight tested concentrations in MCF-7 cells

**D** Violin plots (robust z-scores, feature-level) of selected reference compounds at eight tested concentrations in different cell lines

**E** Evaluation of CPP assay robustness

**F** Reproducibility of robust z-scores in MCF-7 cells

Fig. 2C), with the Mito activity profiles of fluphenazine and tetrandrine showing higher robust z-score magnitudes in CP than in CPP (Fig. 2C, D, Supplementary Fig. 2C). In addition to the observed Lyso-related MoA of the two compounds (see Fig. 2A), those Mito activities were also consistent with published CP profiles of MCF-7 cells[17]. Notably, tetrandrine also induced prominent responses in the RNA/ER and AGP channels in CP, which could be more specifically allocated to

the ER and Golgi channels, respectively, when using CPP (Fig. 2D). This observation is one example of the usefulness of separating the ER and RNA channels, which provides qualitatively different information. This is further supported by the direct comparison of the features extracted for all compounds from only the cytoplasmic region of the RNA and ER channels (CPP) with the merged RNA/ER channel (CP) (Supplementary Fig. 2C). Together, these examples illustrate that CPP expands the

**Fig. 2 | Barcoding compound effects across different cell lines using CPP.**
**A** Reference compounds to benchmark CPP including maximum exposure concentrations and available information on compound type and MoA[17,31–44].
**B** Schematic workflow outlining the CPP image and data analysis procedures. Details about the CP and CPP image analysis pipelines (Harmony) are included in the Supplementary Data 6, 7. **C** Compound activity profiles showing activities of all reference compounds at eight tested concentrations in MCF-7 cells. Heatmaps visualize robust z-scores (median of all $N_{Tech} = 3$ and $N_{Biol} = 4$) for each feature (feature-level) extracted from CPP and CP images captured at 20x magnification. Features are ordered from left to right according to imaging channels (colored bars), with generic (general morphology not related to a specific channel) features (grey bar) shown first. Grey triangles indicate exposure concentration ranges in decreasing order from top (high) to bottom (low), with magenta part of triangles indicating cytotoxic ranges. Detailed plots showing relative cell numbers and cytotoxic concentration ranges observed for each reference compound and cell line are included in the Supplementary Data 8–11. Note that the compound activity profile of tetrandrine indicated insufficient exposure levels at the highest compound concentration. High-resolution versions of the heatmaps are included in the Supplementary Data 12–15. **D** Violin plots showing activities (distribution of robust z-scores of all features per imaging channel) of selected reference compounds at eight tested concentrations in selected cell lines. **E** Layout of technical (TRep) and biological (BRep) replicate plates used in the small-scale reference compound screen for CPP and CP across all cell lines. **F** Pearson correlation matrices for MCF-7 cells comparing intra-plate/technical and inter-plate/biological variability of robust z-score data (at non-cytotoxic concentrations) for CPP and CP images taken at 20x and 40x magnification. The non-cytotoxic concentrations identified for 20x magnification were also applied to the 40x magnification for direct comparisons of the data. AllRep median of all $N_{Tech} = 3$ and $N_{Biol} = 4$; BRep median of all $N_{Tech} = 3$ for each biological replicate, TRep Median of all $N_{Biol} = 4$ for each technical replicate.

multiplexing capacity of the original CP method and, importantly, expands the diversity of the phenotypic profiles.

## CPP is a robust assay for generating reproducible phenotypic profiles

To assess the robustness of the CPP assay, we analyzed the reproducibility of the robust z-scores that were determined for the three intra-plate/technical (TRep) and the four inter-plate/biological (BRep) replicates in the CPP reference compound screen (Fig. 2E). All cell lines generally showed high Pearson correlation scores in the CPP assay for 20x or 40x magnification, which were similar to the Pearson correlation scores obtained when using the CP method in MCF-7 cells (Fig. 2F; Supplementary Fig. 3A). To directly compare the total variabilities of CPP across the four cell lines and between CPP and CP in MCF-7 cells, we further summed up the relative differences of the robust z-scores of all features for each individual TRep, BRep, and AllRep (Supplementary Data 4). The total variabilities between all replicate experiments of each cell line were similar. Those results were consistent with published CP performance metrics[19]. Notably, in MCF-7 cells, the CPP assay showed overall smaller total variabilities compared to the CP method. Our CP and CPP analyses further show that the coefficient of variation of feature data between cells depends mainly on the number of cells being imaged but not on the number of imaging fields per se, indicating that capturing ~ 2500 cells is a sufficient number to ensure statistical robustness of the data (Supplementary Fig. 1F). In conclusion, running the assay in single technical but at least four biological replicates using 20x magnification was also preferred in CPP to ensure its robustness, reproducibility, and applicability to HTPP platforms.

## BMC modeling enables investigation of concentration-dependent phenotypic responses

To determine the specific concentrations at which the reference compounds elicited phenotypic responses, we used the benchmark concentration (BMC) modeling approach as previously described for CP[8,11,12,17]. For each feature, the individual BMC corresponded to the concentration at which the phenotypic response exceeded the defined benchmark response (BMR) cutoff as described before[48]. For example, fulvestrant treatment induced clear concentration-dependent effects for specific Mito channel features (Fig. 3A). Again, the Pearson correlation scores of the BMC values obtained for CPP and CP in MCF-7 cells were very similar, with all cell lines generally showing lower Pearson correlation scores for the BMC values than the robust z-scores (Fig. 2F; Fig. 3C; Supplementary Fig. 3A, B), indicating that the BMC curve fitting increased data variability to some extent. To facilitate analysis of the BMC modeling data, we further assigned the individual features to biologically meaningful feature categories representing specific combinations of channels, cell regions, and analysis modules (Supplementary Data 5) in a similar way as previously described for CP[8,11,12,17].

Using the BMC data and the feature category assignments, we visualized the relative number (proportion) of features that showed significant responses in each feature category. Those Proportion BMC profiles enabled direct comparison of the relative compound activities within and between the four cell lines (Fig. 3B, Supplementary Fig. 3C). Just as described for the compound activity profiles (Supplementary Fig. 2C), the Proportion BMC profiles were generally similar between CPP and CP (Fig. 3B). Notably, this comparison at the feature category level was hampered due to the differences in the number of imaging channels and number/type of extracted features. Several compounds such as rotenone showed broad activities across all feature categories and cell lines, with a large number of single features showing a significant response particularly in MCF-7 and RPTEC cells. Other compounds also showed broad but more cell-line specific activities. For example, the cytostatic drug etoposide showed activities only in proliferating HepG2, U2OS, and MCF-7 cancer cell lines but not in the postmitotic, differentiated RPTEC primary kidney cells, which is in line with studies showing that etoposide sensitivity decreases with cellular differentiation[49]. Notably, treatment with the cytostatic drug fulvestrant resulted in phenotypic responses exclusively in MCF-7 cells reflecting its function as a specific inhibitor of the estrogen receptor alpha signaling pathway, which is a specific trait of the MCF-7 breast cancer cell line[50]. In addition to the described broad activities, the Proportion BMC profiles also enabled the identification of compounds showing activities that were more specific to different feature categories. For example, treatment with siramesine and tetrandrine caused pronounced responses in Lyso-related feature categories in MCF-7, HepG2, and U2OS cells, consistent with the activities described for these compounds[41,42]. These data illustrate that the visualization of relative compound activity profiles using biologically meaningful feature categories can support the characterization of the specific MoA of compounds.

To further distinguish between low-concentration (most sensitive, potentially primary) and high-concentration (less sensitive, potentially secondary) compound effects, we further investigated the specific sequence of feature category responses as well as the maximum sizes of those effects for CPP (Fig. 3D; Supplementary Fig. 4A–O) and CP (Supplementary Fig. 5A–O) using accumulation and magnitude plots as previously described for CP[8,11,12,17]. As illustrated by the example of fulvestrant (Fig. 3D), each data point in the BMC accumulation plot represents a different feature category in a ranked manner to enable the straightforward identification of sensitive feature categories that showed responses at lower concentrations. The corresponding BMC magnitude plot provides more detailed information about the BMCs and the effect sizes for all individual features of those feature categories that were displayed in the accumulation plot. For fulvestrant-treated MCF-7 cells, the BMC accumulation plot of CPP revealed concentration-dependent responses for 56 of the 62 feature categories, concordant with its broad activity shown in the Proportion

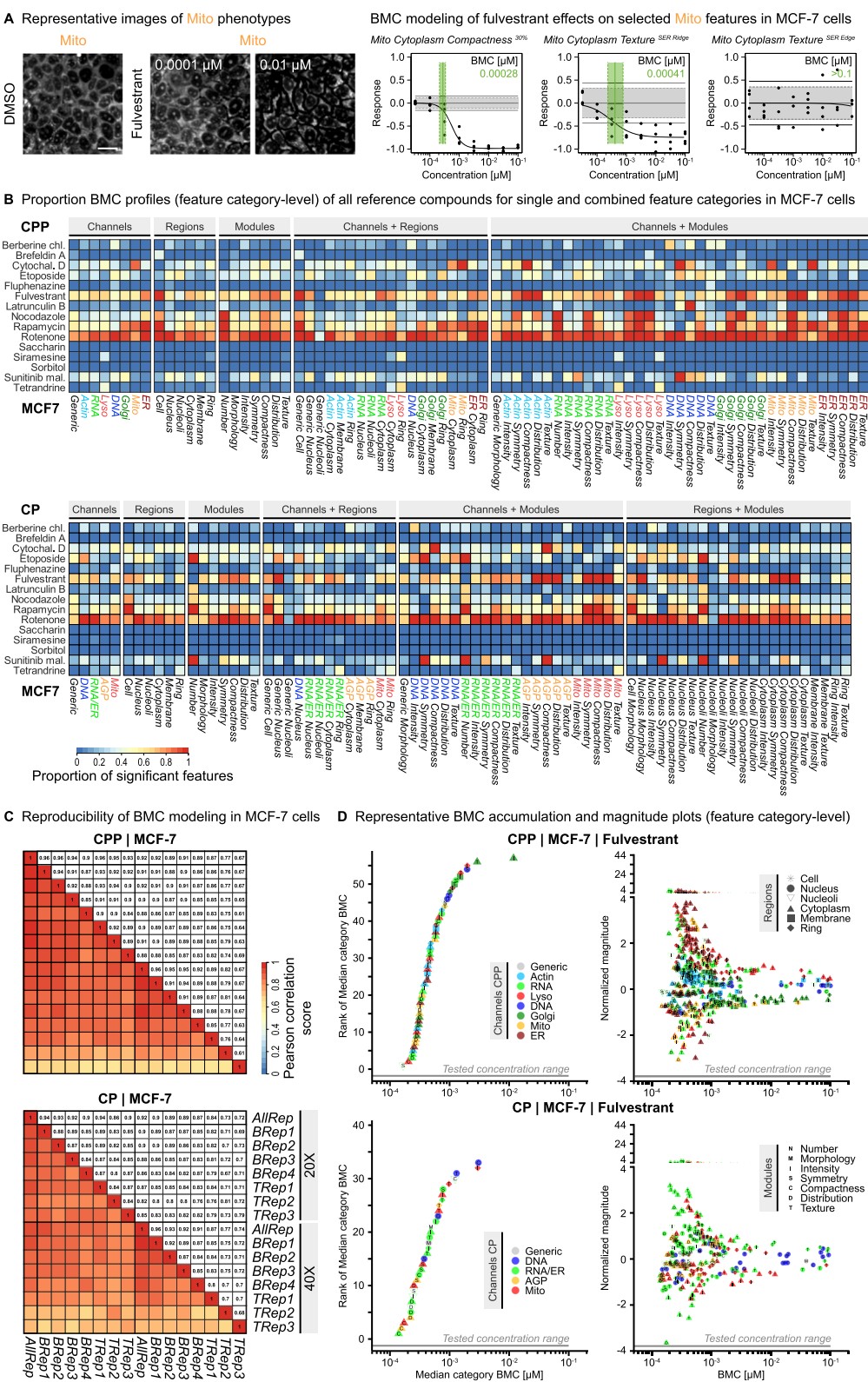

**A** Representative images of Mito phenotypes

BMC modeling of fulvestrant effects on selected Mito features in MCF-7 cells

**B** Proportion BMC profiles (feature category-level) of all reference compounds for single and combined feature categories in MCF-7 cells

**C** Reproducibility of BMC modeling in MCF-7 cells

**D** Representative BMC accumulation and magnitude plots (feature category-level)

BMC profile (Fig. 3B). In particular, CPP was able to distinguish between RNA and ER features showing maximum effect sizes, highlighting the advantage of separating RNA and ER channels to more precisely determine different MoA. The following three use cases describe some examples of applying the CPP assay and the described data analysis, visualization, and interpretation approaches to investigate specific research questions, thereby illustrating its added value.

## Use case 1: Identifying changes in number and morphology of nucleoli

Nucleoli are dynamic nuclear condensates that play a pivotal role in ribosome biogenesis and serve as important stress sensors that, when disrupted, trigger p53-dependent cell cycle arrest[51]. Changes in nucleoli number and/or morphology are associated with cancer, neurodegenerative disorders, and aging, thereby providing a relevant

**Fig. 3 | Investigating substance specific effects using benchmark concentration (BMC) modeling. A** Representative images and BMC fitting curves showing fulvestrant effects on exemplary Mito features in MCF-7 cells. Each data point ($N_{Biol} = 4$) represents a median of robust z-scores from $N_{Tech} = 3$. The light grey area shows the baseline activity (DMSO) ± SD. The horizontal solid black lines show the benchmark response (BMR) cutoff, with $BMR = 1.349*SD$ (according to[48]) and $SD = 1.4826*MAD$ of the corresponding DMSO solvent control wells. The feature BMC is the concentration at which the fitted curve intersects the specific BMR. The light green area indicates the uncertainty range of the determined BMC value. Scale bars = 20 µm. **B** Proportion BMC profiles showing relative activities of all reference compounds in MCF-7 cells. Heatmaps visualize the proportion of single features (from CPP and CP, 20x magnification) of a specific feature category that show significant ($p < 0.05$, one-sided Student's t-test) responses (i.e., low feature BMCs) to treatment with a particular reference compound compared to all compounds

that were tested. The regions + modules panel of CPP is shown in Fig. 4B. Determination of single feature BMCs from $N_{Tech} = 3$ and $N_{Biol} = 4$ as described in (**A**). **C** Pearson correlation matrices for MCF-7 cells comparing intra-plate/technical and inter-plate/biological variability of BMC data as described in Fig. 2F. Note, $BMC^{NA}$ values were set to the specific maximum non-toxic concentration of each compound. **D** Representative BMC accumulation plot and magnitude plots showing the concentration-dependent sequence (rank) and maximum effect size (normalized magnitude, i.e., the maximum robust z-score) of fulvestrant effects on feature categories and single features (from CPP and CP, 20x magnification) in MCF-7 cells, respectively. Each feature category BMC in the BMC accumulation plot is the median of all single feature BMCs of the same category. Determination of single feature BMCs from $N_{Tech} = 3$ and $N_{Biol} = 4$ as described in (**A**). Feature categories for which less than 30% of the included features show a response are excluded from the plots.

biomarker and therapeutic target[52]. Although being visualized by CP in the RNA channel, nucleoli have not yet been directly analyzed as a distinct cellular compartment in CP. Therefore, nucleoli were added in this study to the CPP and CP image and data analysis pipelines to profile potential compound effects on nucleoli-related features (Supplementary Data 3, Supplementary Data 5) such as changes in nucleoli numbers, which already served in genetic screens as a readout to identify regulators of ribosome biogenesis and cell cycle progression[53,54].

In CPP, nucleoli were clearly detectable in the RNA channel based on their spot-like structure in the nucleus region (Fig. 4A), with MCF-7 and RPTEC exhibiting ~ 2 ± 1, HepG2 ~ 3 ± 2, and U2OS ~ 4 ± 2 nucleoli on average per cell. Rapamycin, a highly specific mTOR inhibitor, caused a strong effect on the number of nucleoli per cell (Fig. 4A; Supplementary Fig. 4I), consistent with the role of the mTOR pathway in controlling cell cycle progression through regulating ribosome biogenesis and nucleoli numbers[52]. The compound activity plots (Fig. 4A) and BMC accumulation plots (Fig. 4C) revealed that rapamycin showed activities already at lowest concentrations (3 nM) among the tested reference compounds in MCF-7, HepG2, and U2OS cells, but interestingly no relevant responses in RPTEC cells. As the lowest tested concentration of rapamycin was still too high for calculating a reliable BMC for the nucleoli number feature category, it was set to the next lower concentration (as described in the methods section). In fact, this suggests that nucleoli responses to rapamycin will be observed even in the sub-nanomolar range, corresponding to the published range of the IC50 value.

Furthermore, several perturbations in cell functions, such as impairment of the actin cytoskeleton or generation of reactive oxygen species (ROS), can cause changes in nucleolar number or morphologies and thus alter nucleolar function or trigger nucleolar stress response mechanisms[55,56]. Indeed, treatment with latrunculin B, a disruptor of the actin cytoskeleton, led to a clear increase of nucleoli numbers in MCF-7 cells but to a decrease in RPTEC cells (Fig. 4A–C, Supplementary Fig. 4G). Other compounds such as the Mito inhibitor rotenone, which led to mitochondria-mediated ROS generation in cell culture models[57,58], caused prominent responses in both number and morphology of nucleoli (Fig. 4A–C, Supplementary Fig. 4J). Particularly in RPTEC and MCF-7 cells, rotenone treatment led to the observation of enlarged nucleoli signals at high but non-cytotoxic concentrations, which actually matched the signals in the DNA channel, indicating a severe disruption of the nucleolus structure.

In summary, addition of nucleoli as a distinct cellular compartment to the CPP and CP image and data analysis pipelines enabled comprehensive phenotypic profiling of compound effects on nucleoli and, thus, may provide relevant insights when studying mechanisms of ribosome biogenesis and the nucleolar stress response in basic research. Considering that increased number and size of nucleoli has been shown to correlate with elevated cancer cell proliferation and

poor prognosis[59], this may further support development of biomarkers and therapeutic intervention in a clinical context.

## Use case 2: Differentiating between Actin and Golgi responses at different effective compound concentrations

The actin cytoskeleton is a dynamic network of bundles of actin filaments (F-actin) that controls the shape and motility of cells but also determines the morphology of cellular organelles such as the Golgi apparatus and mitochondria[60–62]. Cytochalasin D (fugal biotoxin) and latrunculin B (marine biotoxin) both inhibit actin polymerization and disrupt actin filament organization, but act through different specific mechanisms[63,64].

In contrast to the CP method, Actin and Golgi signals were captured in separate channels in CPP, which enabled the differentiation between Actin and Golgi responses of cells to compound treatment. In untreated cells, phalloidin staining revealed the characteristic network of actin filaments, e.g., with prominent cortical F-actin portions at the plasma membrane in MCF-7 cells or F-actin bundles forming stress fibers in U2OS and RPTEC cells (Fig. 5A). The corresponding Golgi staining showed the typical compact Golgi structures in close proximity to the nucleus in all cell lines, with some weaker signals at the plasma membrane. Upon exposure to cytochalasin D (Fig. 5A) and latrunculin B (Supplementary Fig. 6A), the actin cytoskeleton collapsed into intensely stained clumps and aggregates, which was accompanied by dispersed fragmentation and clumping of Golgi structures. Notably, these phenotypic changes occurred at different concentrations in cytochalasin D treated cells, with Actin features showing responses at lower concentrations than Golgi features, which was particularly evident in U2OS cells (Fig. 5B, Supplementary Data 23). In cells treated with latrunculin B, separation of Actin and Golgi effects were not that pronounced and generally occurred at higher concentrations (around 1 µM) as compared to cytochalasin D (around 0.1–0.3 µM) (Supplementary Fig. 6B, Supplementary Data 23). The BMC accumulation and magnitude plots further indicated a higher potency of cytochalasin D than latrunculin B with regard to the onset and the maximum effect size of the observed phenotypic changes of Actin and Golgi. Concordant with the essential role of the actin cytoskeleton in mitochondrial fusion and fission[61], the BMC plots also revealed clear effects of cytochalasin D and latrunculin B on Mito features in both CPP (Fig. 5B; Supplementary Fig. 6B) and CP (Supplementary Fig. 5C, G). Importantly, when using the CP method, the BMC plots confirmed the effects of cytochalasin D and latrunculin B on the AGP channel (Fig. 5B, Supplementary Fig. 5G), with the caveat that differentiation between Actin and Golgi responses at different concentrations was not possible. This caveat was also evident when comparing the total number of BMC features responding to cytochalasin D and latrunculin B treatment in either the AGP channel (CP) or Actin and Golgi (CPP) in the BMC bar plots (Supplementary Data 23). Therefore, the specific sequence of these concentration-dependent responses to compound treatment that can be distinguished with the

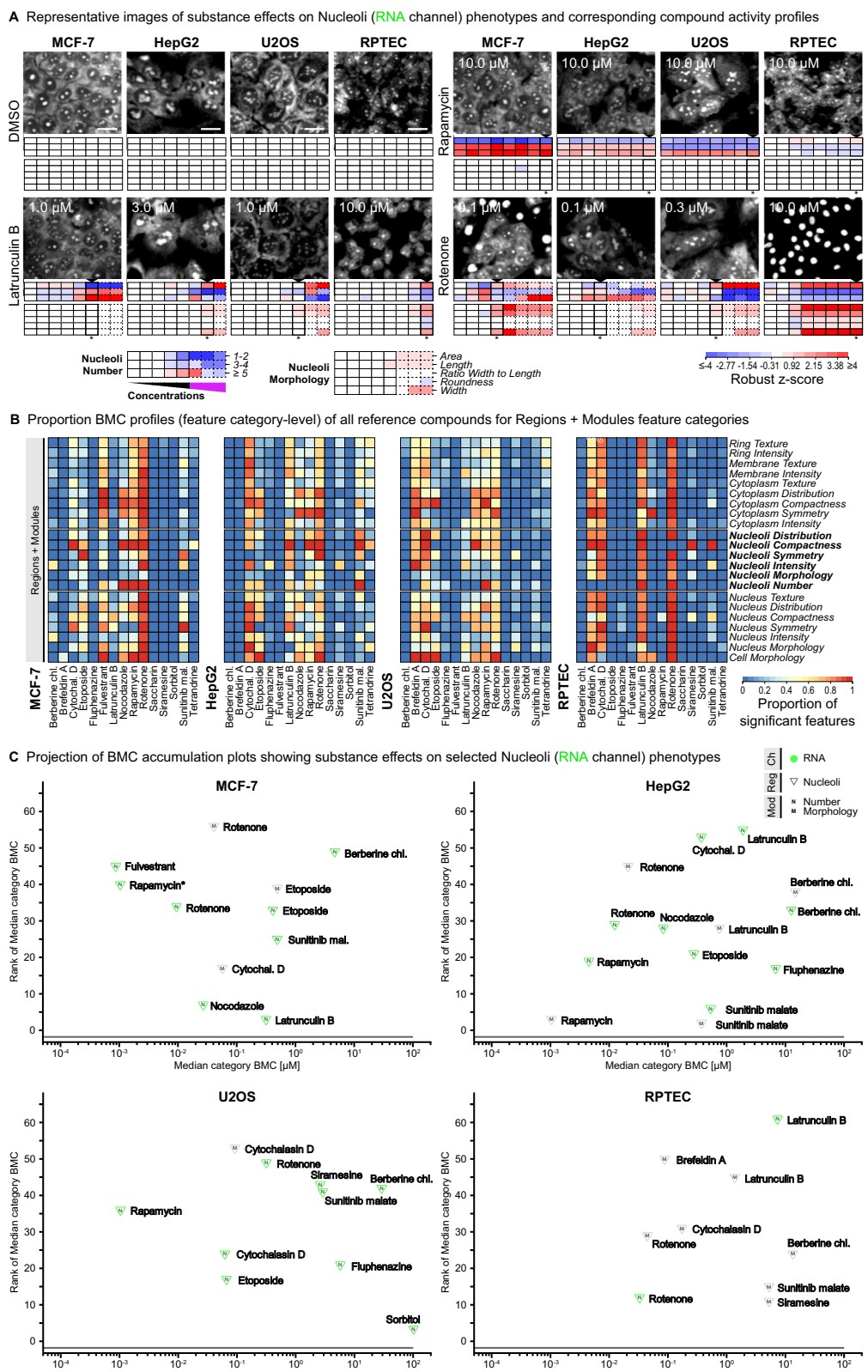

**A** Representative images of substance effects on Nucleoli (RNA channel) phenotypes and corresponding compound activity profiles

**B** Proportion BMC profiles (feature category-level) of all reference compounds for Regions + Modules feature categories

**C** Projection of BMC accumulation plots showing substance effects on selected Nucleoli (RNA channel) phenotypes

CPP assay suggest that perturbation of the actin cytoskeleton, e.g., caused by cytochalasin D, represents an event at lower concentrations leading to subsequent secondary effects on other subcellular compartments and organelles such as Golgi and Mito.

In order to identify reference compounds with activity profiles similar to cytochalasin D and latrunculin B, we used Spearman correlation and hierarchical clustering to generate profile similarity plots for each cell line (Fig. 5C, Supplementary Fig. 6C). These plots show the overall similarity of compound effects (robust z-score, feature level, see Fig. 2C, Supplementary Fig. 2B) at each highest non-cytotoxic concentration. To enable direct comparison of the compound activity profiles of CPP/Harmony, CP/Harmony, and CPP/Cell Profiler in MCF-7 cells, the Lyso features were excluded from this analysis as Lyso was not part of the CP dye set. In MCF-7 cells, the active reference

**Fig. 4 | Identifying changes in number and morphology of nucleoli.**
**A** Representative images (RNA channel) and corresponding compound activity profiles for nucleoli-related features (part of profiles described in Fig. 2C and Fig. S2B) showing activities of four exemplary reference compounds on the two feature categories *RNA Nucleoli Number* (comprising three features) and *RNA Nucleoli Morphology* (comprising five features) at eight tested concentrations (increasing from left to right, dashed outlines indicating cytotoxic ranges) across four different cell lines. Representative images for highest non-cytotoxic concentrations (as indicated by an asterisk below the corresponding concentration under each heatmap profile) are shown and compared to the DMSO solvent control. $N_{Biol} = 4$. Scale bars = 20 µm. **B** Proportion BMC profiles showing relative activities of all reference compounds for regions + modules feature categories across four different cell lines as described in Fig. 3B. **C** Projection of all BMC accumulation plots using Median category BMC and ranks from 15 reference compounds (Supplementary Fig. 4A–O) showing activities on the *RNA Nucleoli Number* (green triangles) and *RNA Nucleoli Morphology* (grey triangles) feature categories across four different cell lines. Color-coding of reference compounds according to the Proportion BMC profiles shown in (**B**).

compounds grouped into three major clusters (#4-6) comprising compounds with annotated Mito-, Lyso-, and Actin-related MoAs (see Fig. 2A, Fig. 5C). The "Mito cluster" (#5) comprised three compounds with the Mito inhibitors berberine chloride and rotenone showing the highest profile similarity in this group (dashed orange squares), which was concordant between CPP/Harmony, CP/Harmony, and CPP/Cell Profiler. In the other cell lines, the two compounds grouped into larger clusters (Supplementary Fig. 6C). The size of the "Lyso cluster" (#6) in MCF-7 cells varied more between CPP/Harmony, CP/Harmony and CPP/Cell Profiler but concordantly included the Lyso modulators fluphenazine and tetrandrine, which showed high profile similarities (dashed red squares). Since Lyso features were excluded from those plots, the clustering of the fluphenazine and tetrandrine was essentially based on their additional activities on ER and Golgi in CPP or RNA/ER and AGP in CP, as detailed in case study 3.

Interestingly, despite their common Actin-related MoA, no clear "Actin cluster" comprising cytochalasin D and latrunculin B was consistently observed in the profile similarity plots of the different cell lines (Fig. 5C, Supplementary Fig. 6C). This indicates that their overall compound activity profiles were rather distinct from each other, which was probably also due to the more pronounced Actin-related responses of cytochalasin D compared to latrunculin B, as shown in the corresponding BMC plots (Supplementary Fig. 5C, G) and the BMC bar plots (Supplementary Data 23). In fact, latrunculin B was part of a major cluster (#4) along with nocodazole and etoposide in both CPP/Harmony and CP/Harmony, and showed a higher correlation to the microtubule inhibitor nocodazole than to cytochalasin D in MCF-7, HepG2 and U2OS cells (dashed cyan squares). In contrast, the clustering of cytochalasin D clearly differed between CPP/Harmony and CP/Harmony in MCF-7 cells. In CP/Harmony, cytochalasin D was part of the "Lyso cluster" (#6), with a high correlation to fluphenazine and tetrandrine that both exerted AGP and prominent RNA/ER responses as shown by the corresponding BMC bar plots (Supplementary Data 23, 24). Similarly, cytochalasin D also showed considerable responses in those two CP channels, which contributed to the observed clustering. When using CPP/Harmony, those responses of fluphenazine and tetrandrine could be more clearly separated into stronger Golgi than Actin and stronger ER than RNA responses (Supplemental material 18 and 19). Importantly, in CPP/Harmony, cytochalasin D showed much stronger Actin than Golgi responses, which contributed to its clear separation from the other compounds in the profile similarity plots line (Fig. 5C), which was also the case in HepG2 and U2OS cells (Supplementary Fig. 6C).

In conclusion, this case study suggests that separating Actin and Golgi channels can be beneficial to enhance the organelle-specificity and diversity of phenotypic profiles for comparative MoA analyses. The example of cytochalasin D and latrunculin B highlights the need for generating phenotypic profiling data across a concentration range and using different analysis methods for detailed MoA analyses. BMC modeling helps to distinguish low-concentration (primary) from higher-concentration (secondary) effects. Investigating maximum effect levels at the highest non-cytotoxic concentrations using profile similarity plots supports clustering of compounds according to overall profile similarities but may miss important responses at different concentrations that contribute to distinct phenotypic profiles.

## Use case 3: Elucidating compound effects on lysosomes

Lysosomes are essential cellular organelles responsible for degradation and recycling of macromolecules and autophagy. They are typically broadly distributed within cells and can form direct contact sites with the ER and mitochondria. The morphology, positioning, motility, and function of lysosomes are closely linked, making them an intensely studied cellular organelle[65]. We therefore included lysosomes as relevant organelles in the CPP assay to enable assessment of compound-induced Lyso phenotypes. We further evaluated the use of siramesine (accumulating in lysosomes leading to lysosome leakage and alterations in lysosomal pH)[41], fluphenazine (involved in hypoxia-induced cell death leading to lysosome aggregation functional impairment)[35], and tetrandrine (lysosomal deacidification agent perturbing autophagic flux)[42] as suitable reference compounds for Lyso phenotypes, which were selected based on their published MoA.

Including Lyso features into the profile similarity plots (Fig. 6A, Supplementary Fig. 7C) further increased the correlation of the three "Lyso cluster" compounds tetrandrine, fluphenazine, and siramesine in all four cell lines as expected. However, in contrast to CPP/Harmony, those three compounds with annotated Lyso-related MoA (see Fig. 2A) did not form a separate "Lyso cluster" in the profile similarity plots of CPP/Cell Profiler, neither for the reduced set of features nor for the complete set (without feature selection applied) (Supplementary Fig. 8C). Interestingly, a high profile similarity to the three compounds was also observed for the multi-targeted tyrosine kinase inhibitor sunitinib malate and particularly evident in MCF-7 and U2OS cells, where only the addition of Lyso features led to the inclusion of sunitinib malate into the "Lyso cluster" in CPP/Harmony but not CPP/Cell Profiler, indicating a similar Lyso-related MoA. Indeed, exposure of MCF-7 cells to sunitinib malate, tetrandrine, fluphenazine, and siramesine led to intensely stained clumps and aggregates in the Lyso channel, accompanied by Golgi and ER disruption (Fig. 6B), which was also observed for the other cell lines (Supplementary Figs. 7A–10A). These pronounced Lyso responses were also evident in the corresponding BMC accumulation and magnitude plots, with primarily texture-related features accompanied with less strong phenotypic responses across different other organelles (Fig. 6C; Supplementary Figs. 7B–10B). Lyso responses to tetrandrine, fluphenazine, and siramesine generally occurred at lower concentrations compared to responses of the other organelles. Notably, this was the opposite in cells exposed to sunitinib malate, indicating that Lyso responses might occur as secondary effects. Moreover, the BMC plots further revealed differences in the magnitude of responses of the four cell lines to compound treatment, with particularly pronounced effects on ER and Golgi (Fig. 6C, Supplementary Figs. 7B–10B). However, when using the CP method, distinction of ER and Golgi responses was not possible from the combined RNA/ER and AGP channels (Fig. 6C, Supplementary Fig. 5E, L, N, O, Supplementary Data 23, 24).

Together, these data demonstrate the capacity of the CPP assay to elucidate compound effects on lysosomes and to differentiate between ER and RNA or Actin and Golgi effects, thereby providing valuable insights into the MoA of compounds. Furthermore, despite their diverse MoA, tetrandrine, fluphenazine, and siramesine induced Lyso phenotypes across all cell lines. Those compounds supported the identification of a Lyso activity of sunitinib malate, which is concordant

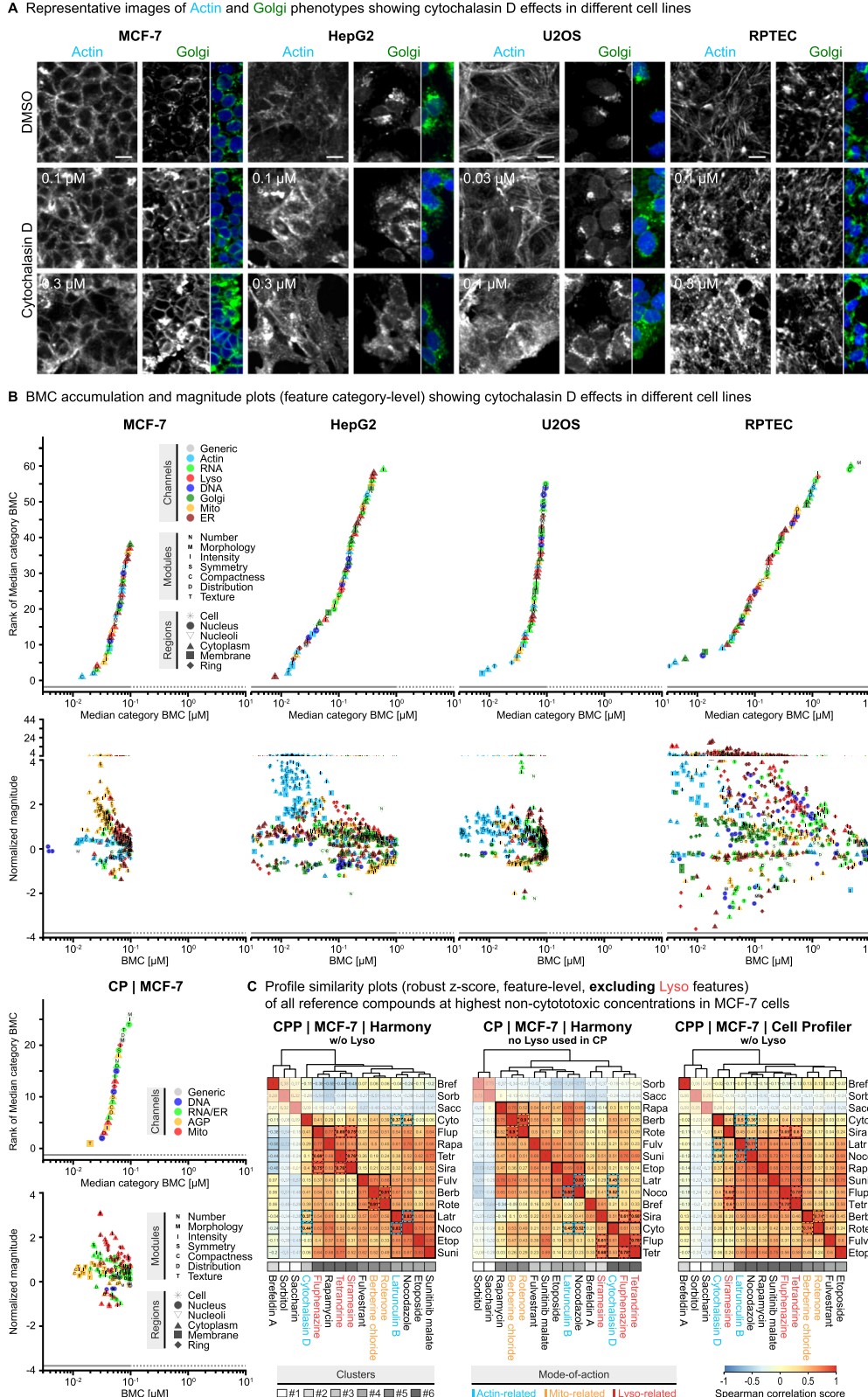

**A** Representative images of Actin and Golgi phenotypes showing cytochalasin D effects in different cell lines

**B** BMC accumulation and magnitude plots (feature category-level) showing cytochalasin D effects in different cell lines

**C** Profile similarity plots (robust z-score, feature-level, **excluding** Lyso features) of all reference compounds at highest non-cytotoxic concentrations in MCF-7 cells

with studies describing its sequestration and accumulation to lysosomes in cancer cells[66,67], indicating their usability as suitable reference compounds in the CPP assay.

## Discussion

In this proof-of-concept study, we established a customizable, robust, and broadly applicable phenotypic screening assay for multiplexed, iterative staining of single cells with at least seven fluorescent dyes labelling at least nine different subcellular compartments and organelles. The CPP method thus complements the versatile HTPP methods and offers additional options for addressing MoA-specific research questions. The conceptual design of the CPP assay was essentially inspired by two well-established approaches enabling multiplexed and spatially resolved phenotypic profiling of single cells – the dye-based

**Fig. 5 | Differentiating between Actin and Golgi responses at different effective compound concentrations using cytochalasin D. A** Representative images (Actin and Golgi channels) showing activities of cytochalasin D on Actin- and Golgi-related features at two non-cytotoxic concentrations compared to the DMSO solvent control across four different cell lines. $N_{Biol} = 4$. Scale bars = 20 μm. **B** Corresponding BMC accumulation and magnitude plots showing cytochalasin D effects on feature categories and single features across four different cell lines in CPP, and MCF7 in CP as described in Fig. 3D. **C** Profile similarity plots showing the correlation of the phenotypic profiles (Spearman correlation of robust z-scores at the feature level, excluding Lyso features) of all reference compounds at each

highest non-cytotoxic concentration in MCF-7 cells. Compounds are assigned to one of six clusters (grey-shaded boxes) based on hierarchical clustering. Compounds are color-coded (cyan, orange, red) according to their annotated Actin-, Mito-, or Lyso-related MoA (see Fig. 2A). Colored, dashed boxes highlight correlation scores of compounds with the same annotated MoA. Input feature data (median of all $N_{Tech} = 3$ and $N_{Biol} = 4$ for each feature) are extracted from CPP (CPP/Harmony, CPP/Cell Profiler) or CP (CP/Harmony) images using Harmony or Cell Profiler image analysis software. Correlation scores of negative control compounds (i.e., saccharine, sorbitol) are shown slightly transparent.

CP method[4–6] and the antibody-based iterative indirect immunofluorescence imaging (4i) method[29]. CPP now combines the advantages of both approaches, i.e., the very efficient staining and analysis of single cells using dyes and standardized feature sets, and the iterative staining and signal removal process allowing multiple rounds of staining on the same sample. In a comparative study using 15 diverse reference compounds targeting relevant organelles and cell functions, we compared the CPP assay with the original CP method using MCF-7 cells to explore its potential to expand the available HTPP methods. Moreover, we investigated cell type- and organelle-specific compound effects using three additional cell lines.

We showed that CPP expands the flexibility, customizability, and multiplexing capacity of the original CP method regarding selection and combination of relevant dyes for the specific research questions and tremendously enriches the information content of morphological profiles. The optimized elution buffer enables the staining, elution, and re-staining of single cells and avoids the current need to swop in, e.g., dyes for lysosomes instead of mitochondria or staining lipid droplets rather than ER[68,69], or to use special synthesized de-stainable phenotyping probes based on with click chemistry reactions[70]. Importantly, subcellular compartment and organelle morphologies were preserved after the elution step reinsuring the reliability of morphological profiles from subsequent staining cycles. Therefore, we assume that the CPP assay can also be performed with multiple rounds of iterative staining-elution cycles, which remains to be experimentally confirmed in follow-up studies.

The phenotypic profiles of CPP and CP and the conclusions drawn were largely consistent with published CP data for MCF-7 cells[17]. However, in contrast to the MitoTracker Deep Red dye used in CP, the MitoTracker Orange dye used in CPP also stained nuclear structures (probably nucleoli) to some extent (Supplementary Fig. 1E). This effect was neither related to the cross excitation of the SYTO 14 dye when using the Mito channel lase line nor a consequence of the elution step as it could also be observed when cells were stained exclusively with the MitoTracker Orange dye. This effect may be attributed to the different properties of the two dyes[71] but did overall not affect the Mito profiles as the nucleus region was not analyzed in the Mito channel in CPP. To mitigate the observed cross excitation of the SYTO 14 dye, which is also still an issue in the current version of the CP assay[5], we tested the usability of the SYTO 9 and SYTO 13 dyes for CPP, which, however, showed too high staining variability.

We further showed that CPP expands the diversity of the phenotypic profiles compared to the original CP method due to the completely separate imaging and analysis of single dyes in individual channels. Although a recently updated version of CP and current technical advancements of high-content imaging systems improve separate imaging of dyes[5,19], the overlapping spectral ranges of co-stained dyes leading to cross-excitation and/or emission bleed-through (at simultaneous excitation) still remain an issue and can affect morphological profile specificity. Therefore, CPP has the capacity to generate more organelle-specific and detailed morphological profiles for more precise determination of different MoA of compounds or genetic perturbations. The case studies indeed suggested that separating Actin and Golgi as well as ER and RNA channels and, in

particular, addition of the Lyso channels may be beneficial to enhance the organelle-specificity and diversity of phenotypic profiles for comparative MoA analyses. However, the number of compounds for different MoA provided in this study was yet limited and the first indications regarding a potential higher phenotypic profile-specificity of CPP over CP will need to be strengthened by comparing the reference compound profiles against a larger set of compounds with known MoA.

Previous reports from several laboratories have also noted the need to perform more extensive filtering of unreliable features when using Cell Profiler[15,23,25]. From our analysis of CPP data, we concur with this observation, because from the large number of features that were initially extracted using Cell Profiler, ~25% of all features had to be immediately filtered out during the data cleaning procedure due to a high number of missing or identical values. The subsequent feature selection process, which is commonly applied to Cell Profiler data[47], further led to the additional filtering of many features with high redundancy or low informative value. Although both CPP/Cell Profiler and CPP/Harmony showed the formation of a "Mito cluster", the three reference compounds with annotated Lyso-related MoA did not form a separate "Lyso cluster" in the profile similarity plots of CPP/Cell Profiler, neither for the reduced set of features nor for the complete set. In that regard, Harmony provided a better separation of the reference compounds. In addition, the required extensive filtering of features made the use of Cell Profiler somewhat less practical overall and may consequently produce different feature lists, which is not optimal when replicate experiments are to be evaluated individually or compared across different screening sites. In turn, when using Harmony, the pre-selected features proved reliable for downstream analysis without the need for additional filtering or feature selection.

In this study, we obtained comprehensive and robust phenotypic profiles when conducting CPP in highly confluent MCF-7 and RPTEC cells as well as sub-confluent U2OS and HepG2 cells (commonly used in CP), which in most cases aligned well with the known MoA of the 15 diverse reference compounds. Therefore, we strongly encourage scientists to perform CP-based methods under more physiologically relevant cell culture conditions, utilizing a greater variety of cell culture conditions, while at the same time accepting potentially less favorable conditions for spatial imaging. This will ultimately improve the physiological relevance of the cellular models and expand the diversity of public phenotypic profiling datasets for the benefit of basic and translational research, drug discovery, and regulatory toxicology.

The high robustness and reproducibility of the CPP assay and its quantitative image as well as data analysis workflows make it readily applicable to automated liquid handling platforms and high-content imaging systems that are available in many academic laboratories or high-throughput screening facilities. Practical considerations for rapid implementation and seamless integration of the CPP assay into existing phenotypic screening routines should include careful optimization and fine-tuning of the assay parameters as exemplified in this study. Even though we did not observe prominent plate effects influencing the robustness of the CPP assay, fully randomized plate layouts would be generally preferable, but require adequate, automated liquid handling routines[11,12,17]. It is further important to consider that, in

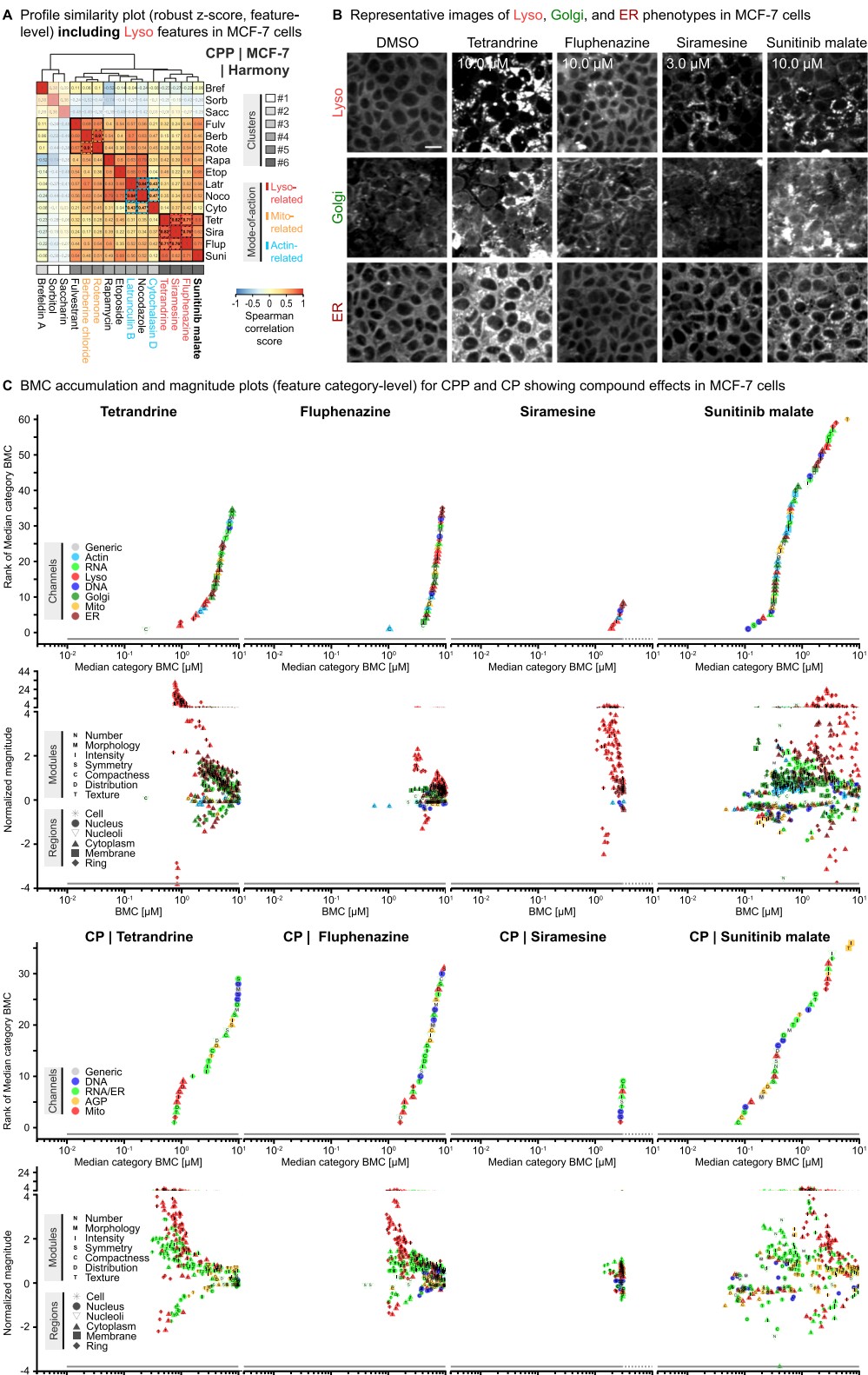

**A** Profile similarity plot (robust z-score, feature-level) **including** Lyso features in MCF-7 cells

**B** Representative images of Lyso, Golgi, and ER phenotypes in MCF-7 cells

**C** BMC accumulation and magnitude plots (feature category-level) for CPP and CP showing compound effects in MCF-7 cells

**Fig. 6 | Elucidating compound effects on lysosomes. A** Profile similarity plots as described in Fig. 5C but including Lyso features. **B** Representative images (Lyso, Golgi, and ER channels) showing activities of four exemplary reference compounds on Lyso-, Golgi-, and ER-related features at a non-cytotoxic concentration compared to the DMSO solvent control across four different cell lines. $N_{Biol} = 4$. Scale bars = 20 μm. **C** Corresponding BMC accumulation and magnitude plots showing the effect of the four exemplary reference compound effects on feature categories and single features in MCF7 cells in CPP and CP as described in Fig. 3D.

comparison to CP, the CPP assay necessitates (at least) one additional elution, staining, and imaging cycle, thereby demanding larger data storage and high-performance computing capacity. However, working with cells at higher confluency would reduce the number of imaging fields, without obvious confounding effects on the statistical robustness of the feature data (Supplementary Fig. 1F), and thus also the required amount of imaging data.

Nevertheless, although CPP offers additional options for more detailed MoA dissection, it has certain limitations compared to CP. With developing the CPP method, we aimed to provide the growing Cell Painting community with a more customizable version of the very successful CP approach with increased organelle-specificity of the phenotypic profiles, accepting the trade-off that this optimization reduces the HTPP capacity of CPP in direct comparison with CP. Since not every research question may require the separation of the AGP and RNA/ER channel or addition of the Lyso channel, it will be up to the individual scientist to decide, if the advantages of CPP serve their specific research needs and outweigh a higher investment in cost, time, and resources compared to CP. In particular, using CPP to extract a greater phenotypic profile diversity from a single cell using iterative staining-elution cycles can be very relevant for research projects using cellular models that are more expensive and laborious than the cancer cell lines commonly used in HTPP. Examples include cellular models that require greater effort for maintenance and differentiation (e.g., iPSC-derived cellular models or neuronal cell lines) or in cases where availability of raw material is limited (e.g., primary cell lines, patient-derived cells). Although such more complex cellular models, and in particular 3D models, have not been widely used in the conventional HTPP studies, they become increasingly important for generation of human-relevant phenotypic profiling data in both pharmaceutical industry and toxicology.

The main scope of this study is to introduce the CPP method using a smaller set of reference compounds with defined MoA, including compounds that were already used before to evaluate the CP method in different cell types[17]. Therefore, the biological insights provided in this study are still very limited but the already ongoing application of CPP in phenotypic profiling consortia, e.g., to identify the biological mechanisms leading to compound-induced cell toxicities and potential hazards of chemicals to human health, will further demonstrate its added value. In addition, the approach taken in this study to evaluate the CPP method using reference compounds at multiple, partially cytotoxic, concentrations aimed at comparing CPP against published CP data. Although this approach can enable the evaluation of the transferability of CPP within and between labs, it was not intended to be transferred to large-scale HTPP of hundreds of thousands of compounds. In fact, depending on the site- and screen-specific requirements, unnecessary costs can be avoided by considering already available compound- and cell type-specific cytotoxicity information or conducting cytotoxicity pre-screens to identify effective but not overly cytotoxic concentration ranges to be used for subsequent phenotypic profiling using CPP. In this way, CPP can be used cost- and time-efficiently for both primary and secondary screening applications.

Future perspectives include the application of the CPP assay for phenotypic screening of larger compound libraries of toxicological and pharmaceutical relevance[9,72–74], allowing the further evaluation of the ability of the CPP assay to generate more comprehensive and MoA-specific morphological profiles as input for virtual screens and compound bioactivity predictions using machine-learning or deep-learning approaches[75]. Practical examples for these computational approaches include matching morphological profiles from compounds to those of genetic perturbations to better understand the genetic pathways affected by the compound at study, relating morphological profiles with structural properties of compounds to predict structure-activity-relationships, or even matching the morphological profiles to entirely orthogonal data from biochemical assays and transcriptomic,

metabolomic, and lipidomic studies to better understand the function of the compounds[20–22,24,26,27,76]. Those data will also provide further mechanistic insights useful for enriching toxicological Adverse Outcome Pathway (AOP) networks for human diseases[77], and enable assessment of the potential use of CPP as a *New Approach Methodology* (NAM), e.g., to predict endocrine active substances (EAS) or specific target organ toxicities (STOT)[78], while avoiding unnecessary animal testing.

## Methods

### Cell lines and routine cell culture
The MCF-7/vBOS (Michigan Cancer Foundation-7/variantBOS) cell line was described before[10,50]. The U2OS (HTB-96) and HepG2 (ACC 180) cell lines were obtained from ATCC and DSMZ, respectively. The RPTEC-TERT1 cell line was obtained from Bob van de Water (Leiden University, The Netherlands)[79]. The identities of all cell lines were verified using the Eurofins Genomics Cell Line Authentication service (Eurofins Genomics, Ebersberg, Germany). Cells were routinely maintained at 37 °C and 5% $CO_2$ over a maximum of 10 passages, and regularly tested using the Eurofins Genomics mycoplasma test service (Eurofins Genomics, Ebersberg, Germany).

MCF-7/vBOS cells were cultured in Dulbecco's modified Eagle's medium (DMEM, 1.0 g/l glucose, sodium pyruvate, no glutamine, no phenol red) (No.11880-028, Lot.2579330, Gibco/Thermo Fisher Scientific, Waltham, MA, USA) supplemented with 5% (v/v) fetal bovine serum (FBS) (No.S0615, Lot.0167 F (estradiol levels: 22.3 pg/ml), Biochrom/Merck, Darmstadt, Germany), 2 mM GlutaMAX (No.35050-061, Lot.2523107, Gibco/Thermo Fisher Scientific), and 100 µg/ml streptomycin / 100 U/ml penicillin (Biochrom/Merck). HepG2 cells were cultured in the same DMEM as MCF-7 cells supplemented with 5% (v/v) FBS (No.BCBQ7890V, Sigma), 2 mM GlutaMAX (No.35050-061, Lot.2523107, Gibco/Thermo Fisher Scientific), and 100 µg/ml streptomycin / 100 U/ml penicillin (Biochrom/Merck). U2OS cells were cultured in DMEM (4.5 g/l glucose, sodium pyruvate, glutamine, no phenol red) (No.P04-03588; Lot.6111121, Gibco/Thermo Fisher Scientific) supplemented with 5% (v/v) FBS (same as for HepG2) and 100 µg/ml streptomycin/100 U/ml penicillin (Biochrom/Merck). RPTEC-TERT1 cells were cultured in DMEM (no glucose, sodium pyruvate, no glutamine) (No.11966025, Gibco/Thermo Fisher Scientific) supplemented with 50% (v/v) Ham's F-12 Nutrient Mix (No.21765029, Gibco/Thermo Fisher Scientific), 2 mM GlutaMAX (No.35050-061, Lot.2523107, Gibco/Thermo Fisher Scientific), 5 µg/ml Insulin, 5 µg/ml Transferrin, 5 ng/ml sodium selenite (No.I1884-1VL, Sigma-Aldrich/Merck), 0.5% (v/v) FBS (No.FBSEU500, Tico Europe, Amstelveen, NL), 10 ng/ml hEGF (No.E9644-2MG, Sigma-Aldrich/Merck), 36 ng/ml Hydrocortisone (No.H0135-1MG, Sigma-Aldrich/Merck), 100 µg/ml streptomycin / 100 U/ml penicillin, and 0.1 mg/ml geneticin (G418) (No.04727894001, Roche, Basel, CH).

The same cell culture conditions were also used in the screening of reference compounds described below. The generation of the reference compound plate, preparation and application of exposure media to cells as well as all staining and elution procedures were performed using a JANUS Automated Liquid Handling Workstation (Revvity Inc., Waltham, MA, USA) and customized protocols written in WinPREP (Revvity Inc.).

### Reference compound plate
All 15 reference compounds (supplier as outlined in Supplementary Data 1, 2) were dissolved in DMSO (Sigma-Aldrich/Merck) at initial stock concentrations (1000X of highest concentration tested) close to the respective solubility limits (≤100 mM). Using an Echo 650 acoustic liquid handler (Beckman Coulter, Brea, CA, USA), eight serial dilutions were generated for each replicate compound at a half-log ratio and added to the 384-well reference compound plate along with 24 DMSO solvent control wells used for definition of baseline activities

(Supplementary Data 2). The individual concentration ranges were defined based on published and internal historical data to achieve relevant exposure conditions leading to detectable changes in phenotypic profiles, while avoiding excessive cytotoxicity at multiple concentrations. This way, the selected concentration ranges included sufficiently high exposure levels to ensure confidence in inactive classifications, but also sufficiently low exposure levels to allow differentiation of early (most sensitive, potentially primary) and later (less sensitive, potentially secondary) responses in cells.

## Cell seeding and compound exposure

For screening of reference compounds, cells from passages 2–4 were seeded in 45 μl culture medium at a density of 7000 MCF-7 cells/well, 5500 HepG2 cells/well, 3000 U2OS cells/well in 384-well PhenoPlates (Revvity Inc.), or 6000 RPTEC cells/well in 384-well Screenstar plates (Greiner Bio-One, Frickenhausen, Germany). For HepG2 and RPTEC cells, 384-well plates were pre-coated with 40 μl/well of 0.1 mg/ml collagen type I (No.C3867-1VL; Lot.SLCN8778, Sigma-Aldrich/Merck) in water for 1 h and then washed once with 40 μl PBS. Cells were grown for 24 h (MCF-7, HepG2, U2OS) or 10 days (RPTEC), with culture medium exchange for RPTEC cells every 2–3 days, followed by compound treatment. At this step, 10 μl of exposure medium containing the reference compounds (each at 5X concentration) was added to the 40 μl culture medium. Cells were then exposed to reference compounds for 48 h with a maximum background DMSO concentration in both compound-treated and solvent control wells of 0.1%.

## Staining procedures

Before preparation of staining solutions, fluorescent dyes or antibodies were thawed at room temperature for 1 h in the dark, then mixed, and briefly (~5–10 sec) centrifuged at maximal speed with a table centrifuge. All aspiration and washing steps described below were performed using a plate washer ELx405 Select CW Microplate Washer (BioTek Instruments, Winooski, VT, USA). If not otherwise stated, media were gently aspirated leaving a minimal residual liquid volume of ~15 μl in each well to avoid drying and disruption of cells. Dyes were obtained from different suppliers as outlined in Supplementary Data 1.

## Cell Painting

The Cell Painting method was generally performed as described in the original or recently updated CP papers[4,5], with adjustments, e.g., of optimal dye concentrations, required to meet lab-specific imaging requirements.

Live-cell staining included aspiration of exposure medium to ~15 μl, addition of 30 μl 1.5X live-cell staining solution containing MitoTracker Deep Red FM in cell culture medium (500.0 nM in staining solution; 333.3 nM in well), and incubation for 30 min at 37 °C and 5% CO$_2$.

Fixation of cells included aspiration of the staining solution to ~15 μl, addition of 30 μl fixation solution containing 4% paraformaldehyde (PFA, in PBS; ~2.7% in well), and incubation for 20 min at room temperature (RT) protected from light.

Permeabilization of cells included complete aspiration of the fixation solution, washing once with 100 μl PBS, addition of 30 μl permeabilization solution containing 0.1% Triton X-100 (in PBS; 0.07% in well), and incubation for 20 min at RT protected from light.

Fixed-cell staining included complete aspiration of the permeabilization solution, washing three times with 100 μl PBS, aspiration to ~15 μl, addition of 30 μl 1.5X fixed cell staining solution containing Hoechst 33342 (4.0 μM in staining solution; 2.7 μM in well), Concanavalin A Alexa Fluor 488 Conjugate (10.0 μg/ml in staining solution; 6.7 μg/ml in well), Wheat Germ Agglutinin Alexa Fluor 555 Conjugate (1.5 μg/ml in staining solution; 1.0 μg/ml in well), Alexa Fluor 568 Phalloidin (33.0 nM in staining solution; 22.0 nM in well), and SYTO 14

Green Fluorescent Nucleic Acid Stain (3.0 μM in staining solution; 2.0 μM in well) (each in PBS containing 1% BSA), and incubation for 30 min at RT protected from light. Finally, plates were washed three times with 100 μl PBS without final aspiration.

## Cell Painting PLUS

**Staining Cycle 1.** Live-cell staining included aspiration of exposure medium to ~15 μl, addition of 15 μl 2X live-cell staining solution containing LysoBrite NIR Fluorescence (1:250 in staining solution (buffer A containing dye); 1:500 in well) and MitoTracker Orange CMTMROS (1.0 μM in staining solution; 0.5 μM in well) (both in LysoBrite NIR Fluorescence buffer B containing 1% BSA; 1:2 in well), and incubation for 30 min at 37 °C and 5% CO$_2$.

Fixation of cells included no aspiration of the staining solution but direct addition of 30 μl 2X fixation solution containing 8% PFA (in PBS; 4% in well), and incubation for 20 min at RT protected from light.

Permeabilization and fixed-cell staining of cells was performed simultaneously and included the complete aspiration of the fixation solution, washing three times with 100 μl PBS, aspiration to ~15 μl, addition of 15 μl 2X permeabilization and fixed-cell staining solution containing Triton X-100 (0.2% in staining solution; 0.1% in well), Alexa Fluor Plus 405 Phalloidin (66.0 nM in staining solution; 33.0 nM in well), and SYTO 14 Green Fluorescent Nucleic Acid Stain (0.6 μM in staining solution; 0.3 μM in well) (in PBS containing 1% BSA; 0.5% in well), and incubation for 30 min at RT protected from light. Finally, plates were washed three times with 100 μl PBS without final aspiration.

**Dye elution.** After imaging, elution of cycle 1 dyes included washing three times with 100 μl deionized water, aspiration to ~15 μl, addition of 40 μl 1.375X elution buffer containing 0.5 M L-Glycine and 1% SDS (pH 2.5) (in deionized water), and incubation for 10 min at RT. The elution buffer was stored at RT and used within one month. Finally, plates were washed three times with 100 μl deionized water and three times with 100 μl PBS without final aspiration. To check for successful elution of cycle 1 dyes, nine random control wells were imaged with identical setting as used in cycle 1.

**Staining cycle 2.** Immediately after elution, the second fixed-cell staining included aspiration of PBS to ~15 μl, addition of 15 μl 2X fixed-cell staining solution containing Hoechst 33342 (8.0 μM in staining solution; 4.0 μM in well), Wheat Germ Agglutinin Alexa Fluor 488 Conjugate (2.5 μg/ml in staining solution; 1.3 μg/ml in well), and Concanavalin A Alexa Fluor 647 Conjugate (20.0 μg/ml in staining solution; 10.0 μg/ml in well) (in PBS containing 1% BSA; 0.5% in well), and incubation for 30 min at RT protected from light. Finally, plates were washed three times with 100 μl PBS without final aspiration.

## Actin staining

Fixation of cells included aspiration of the staining solution to ~15 μl, addition of 15 μl fixation solution containing 8% PFA (in PBS, 4% in well), and incubation for 20 min at RT.

Permeabilization of cells included complete aspiration of the fixation solution, washing three times with 100 μl PBS, aspiration to ~15 μl, addition of 15 μl permeabilization solution containing 0.2% Triton X-100 (in PBS, 0.1% in well), and incubation for 20 min at RT.

Blocking of cells included complete aspiration of the permeabilization solution, washing three times with 100 μl PBS, aspiration to ~15 μl, addition of 15 μl blocking solution containing 2% BSA (in PBS, 1% in well), and incubation for 1 h at RT.

Antibody staining included aspiration of the blocking solution to ~15 μl, addition of 15 μl 2X antibody staining solution containing rabbit anti-Actin [EPR16769] directly-conjugated to Alexa Fluor 555 (No.ab208080, Abcam, Cambridge, UK) (0.5 μg/ml in staining solution; 0.25 μg/ml in well) (in PBS containing 1% BSA in well; 0.5% in well), and

incubation over night at 4 °C protected from light, and washing three times with 100 μl PBS without final aspiration.

Subsequent dye staining included aspiration of PBS to -15 μl, addition of 15 μl 2X fixed-cell staining solution containing Hoechst 33342 (2.8 μM in staining solution; 1.4 μM in well), Alexa Fluor Plus 405 Phalloidin (66.0 nM in staining solution; 33.0 nM in well), Phalloidin-FITC (No.P5282, Sigma-Aldrich/Merck) (0.25 μg/ml; 0.125 μg/ml in well), Phalloidin-iFluor647 (No.ab176759, Abcam, Cambridge, UK) (0.1X; 0.05X in well), and Phalloidin-AlexaFluor647 (No.A22287, Thermo Fisher Scientific) (82.5 nM; 41.25 nM in well) (in PBS containing 1% BSA; 0.5% in well), and incubation for 10 min at RT protected from light. Finally, plates were washed three times with 100 μl PBS without final aspiration.

### Imaging
If not otherwise stated, all dyes were sequentially imaged at an Opera Phenix High-Content Screening System (Revvity Inc.) (4 excitation laser, 4 emission filters) in confocal mode and 2 × 2 pixel binning using a 20x water objective (NA 1.0) or 40x water objective (NA 1.1). Images were taken at two (MCF7), three (RPTEC), or five (HepG2/U2OS) standardized fields/well for 20x or 5 fields/well (MCF-7) for 40x magnification and two (20x) or three (40x) optical sections with 2–4 μm (20x) or 2 μm (40x) spacing between imaging planes, if not otherwise stated. Further details about the microscope setup and imaging settings are summarized in Supplementary Data 1.

### Image analysis and feature extraction
**Harmony.** Image analysis was performed on single image planes using the integrated Harmony software [v4.8] (Revvity Inc.) and customized image analysis sequences for CPP and CP (Supplementary Data 6, 7). Illumination correction preprocessing of raw images was performed in Harmony using the advanced flatfield correction mode. For analysis of CPP data, an application-specific building block (ABB), *Add Channels 4i*, was developed in co-operation with Revvity Inc., which combined cycle 1 and cycle 2 images into a single multi-channel stack based on the Mito signal that is present in both cycles. The ABB calculated a cross correlation between the two Mito images to determine the lateral shifts between the two measurements of cycle 1 and 2. The lateral shift was then applied to all images from the cycle 2 measurement. The correction model that was used in *Add Channels 4i* has a narrow acceptance range for parameters. A maximum shift of 50 pixel in x and y direction (<5% of image width) is assumed, and no scaling nor rotations are expected. In performance tests, the correlation was always sufficiently good that no manual check was deemed necessary. To retain the same image size, the not covered area was filled with zero value pixels. The ABB also shifted a provided mask of the whole valid area from the cycle 1 measurement to determine the area in the image, where all measurements had valid image data, which was then used for further image analysis.

The subsequent steps of the image analysis sequence were similar for both CPP and CP, with some necessary modifications made to the previously published image analysis sequence for CP[11]. Briefly, nuclei were segmented using the Hoechst 33342 channel to define each object. Next, cell outlines were identified using the RNA/ER channel images (CP) or a merged RNA+Mito$^{cycle1}$+Mito$^{cycle2}$ channel image (CPP). Border cells touching the edge of the image were excluded from further analysis. From these nuclei and cell segmentation masks, additional cell regions were defined: cytoplasm, nucleoli, ring, membrane (Supplementary Fig. 2A). For all relevant combinations of channels and cell regions, morphological (Standard, STAR), texture (SER, Haralick, Gabor), intensity, shape, and position features were extracted. Instead of Mito$^{cycle1}$, the Mito$^{cycle2}$ staining was chosen for feature extraction to avoid potential cross-excitation from the RNA channel (SYTO 14). Finally, single-cell data were exported and fed into downstream data analysis workflows using the KNIME software. The individual Harmony sequences for CP and CPP image analysis are available as

Supplementary Data 19 and publicly available on Zenodo (https://doi.org/10.5281/zenodo.14982928). The *Add Channels 4i* ABB [v1.0.4] is available on Zenodo (https://doi.org/10.5281/zenodo.15119993).

**Cell profiler.** To enable analysis of CPP image data using the widely used open-source Cell Profiler image analysis software [v4.2.4][45], all raw images from the two staining cycles were first exported from Harmony and then subjected to illumination correction preprocessing using the Cell Profiler illumination correction pipeline modified from the previously published pipeline for CP[4]. For subsequent combination of cycle 1 and cycle 2 image stacks (as described for the Harmony ABB), we developed the customized Python software *4i stitcher*, which employed a template-matching algorithm based on a normalized cross-correlation coefficient, permitting translational shifts up to 50% of the image width, ± 5% scaling, and ± 5 degrees rotation. Registrations with a cross-correlation coefficient below 0.8 underwent manual verification for accuracy, and those below 0.5 were excluded. Following registration, images from all channels and runs were trimmed to align with the overlapping regions. To enhance throughput, the *4i stitcher* processed images in parallel. In addition to image registration, the *4i stitcher* further generated a merged channel image overlaying all available channels in pseudo-color. For future development, this python-based code can be easily adopted for different microscope systems and/or extended for more advanced (pre-)processing steps. To enable adoption of the methodology, the *4i stitcher* software [v1] is publicly available on Zenodo (https://doi.org/10.5281/zenodo.13784742).

CPP image analysis was performed using a modified CP pipeline previously published[4]. The illumination corrected and registered CPP images, were imported into Cell Profiler for segmentation of nuclei (Hoechst 33342 channel), cell outlines (merged channel) and definition of the other cell regions as described for the Harmony software.

Nucleoli were defined using a speckles module, and features encompassing morphological, distribution, texture, intensity, and granularity were extracted from relevant combinations of channels and cell regions. Due to the multiple image export/import steps required for illumination correction, image registration, and the final image analysis, using Cell Profiler for analysis of CPP images is more time-consuming than using Harmony in our hands. Therefore, Cell Profiler was only used for the analysis of CPP images from MCF-7 cells at 20x magnification to enable comparison of image analysis results with Harmony and evaluate cross-platform compatibility of CPP. The individual Cell Profiler sequences, e.g., for illumination correction and image analysis, are available as Supplementary Data 20 and publicly available on Zenodo (https://doi.org/10.5281/zenodo.14982928).

### Data analysis
The open-source process automation software KNIME [v4.7.1][46] was used to build multiple customized pipelines for image data processing, evaluation and visualization, which generally followed the procedures previously described for CP data analysis using the R software [v4.2.2][8,12], with some adjustments as indicated below. Although imaging-based profiling data comprises features that are inherently correlated, available feature reduction and feature selection strategies have not been conducted (except for Cell Profiler data) in this study to maintain full comparability of the data generated between different screening sites in the context of transferability studies. The individual KNIME pipelines, e.g., for data standardization, generation of compound activity profiles, evaluation of phenotypic profile variabilities, BMC modeling, generation of accumulation and magnitude plots, are available as Supplementary Data 21 and publicly available on Zenodo (https://doi.org/10.5281/zenodo.14982928). Using the unprocessed CP/Harmony and CPP/Harmony features, the influence of the number of analyzed cells and imaging fields on the variance of feature data was evaluated using the coefficient of variation (CV). For this, the predefined number of cells were randomly chosen with repetition for the

field/s of view considering each well and plate independently. Next, the median of each feature among the chosen cells and their CV was calculated per well across plates. The median of those CVs was plotted depending on the pre-defined number of cells used in Supplementary Fig. 1F.

**Standardization.** After data cleaning and assignment of wells to treatment conditions, all extracted features of each individual cell were standardized to the median and median absolute deviation (MAD) of the pooled DMSO solvent control cell population from the corresponding plate (24 wells/plate) to derive for each feature a robust z-score per cell ($Robust\,z\text{-}score_{cell}^{feature}$) according to Eq. 1.

$$Robust\,z\text{-}score_{cell}^{feature} = \frac{X_{\exp/ctrl\,cell}^{feature} - Median_{ctrl\,cells}^{feature}}{1.4826 * MAD_{ctrl\,cells}^{feature}} \quad (1)$$

$X_{\exp/ctrl\,cell}^{feature}$ = Value of a given feature from a single experimental (exp) or solvent control (ctrl) cell;

$Median_{ctrl\,cells}^{feature}$ and $MAD_{ctrl\,cells}^{feature}$ = Median and MAD of a given feature from <u>all</u> pooled solvent control (ctrl) cells (24 solvent control wells per plate);

1.4826 = Constant scaling factor used as approximation for 1 SD of a Gaussian distributed data set

Features with a DMSO MAD of zero were removed from the final feature list (this was only necessary for Cell Profiler data). Next, those cell-level feature data were aggregated to the well-level by calculating for each feature the median of cell-level robust z-scores per well ($Median\,robust\,z\text{-}score_{well}^{feature}$, equation 2), without further standardization (e.g., scaling to the SD or MAD of solvent control wells). Specifically, for CPP/Cell Profiler, standardization of cell features was followed by an additional step of feature selection/reduction using Pycytominer to align with common practice when using the Cell Profiler software for image analysis[47]. For generation of the compound similarity plots shown in Supplementary Fig. 8C, this feature selection/reduction step was performed for the Cell Profiler feature lists either including or excluding Lyso features.

Importantly, all features characterizing cell populations (wells) but no single cells were only standardized at the well-level according to Eq. 3. This applied to the total number of cells per well (*AbsoluteCellNumber*) and the three nucleoli-related *RNA* features *RelativeCellNumber(1-2Nucleoli)*, *RelativeCellNumber(3-4Nucleoli)*, and *RelativeCellNumber(5+Nucleoli)* of the *RNA Nucleoli Number* feature category. Moreover, those three nucleoli-related *RNA* features were calculated using an unmodified z-score due to their discrete values.

$$Robust\,z\text{-}score_{well}^{feature} = \frac{X_{\exp/ctrl\,well}^{feature} - Median_{ctrl\,wells}^{feature}}{1.4826 * MAD_{ctrl\,wells}^{feature}}$$
$$OR\quad z\text{-}score_{well}^{feature} = \frac{X_{\exp/ctrl\,well}^{feature} - Mean_{ctrl\,wells}^{feature}}{SD_{ctrl\,wells}^{feature}} \quad (3)$$

$X_{\exp/ctrl\,well}^{feature}$ = Value of a given feature from a single experimental (exp) or solvent control (ctrl) well;

$Median_{ctrl\,wells}^{feature}$ OR $Mean_{ctrl\,wells}^{feature}$ and $MAD_{ctrl\,wells}^{feature}$ OR $SD_{ctrl\,wells}^{feature}$ = Median OR Mean and MAD OR SD of a given feature from <u>all</u> pooled solvent control (ctrl) wells (24 solvent control wells per plate);

1.4826 = Constant scaling factor used as approximation for 1 SD of a Gaussian distributed data set

Those well-level robust z-scores (from *equation 2* and Eq. 3) were finally aggregated as median over technical and then biological replicates and plotted as heatmaps to generate compound activity profiles of all features per compound and concentration (Supplementary Data 3) for all replicate (*AllRep*) replicate experiments conducted for CPP and CP at 20x and 40x magnification.

**Identification of cytotoxic concentration ranges.** The standardization workflow also included the identification of cytotoxic concentration ranges. Cytotoxic concentrations were defined as those that led to a reduction of the relative number of cells per well and biological replicate (*PercentCells*) below 53% compared to the DMSO solvent control, resulting in a cytotoxicity threshold of 53% for CP/Harmony and CPP/Harmony at 20x magnification. The non-cytotoxic concentrations that were identified for the reference compounds at 20x magnification (based on the 53% threshold), were also applied to the 40x magnification in order to enable direct comparisons of the CPP/Harmony data obtained at different magnifications. Cytotoxic concentrations of compounds from the CPP/Cell Profiler analysis were also aligned to CPP/Harmony to support their direct comparison.

**Evaluation of phenotypic profile variability.** The CPP data analysis workflow included a quality control step, in which the intra-and inter-plate phenotypic profile variabilities were determined for each replicate experiment. For each feature, the differences of the well-level robust z-scores (from *equation 2* and Eq. 3) between a single replicate well and the corresponding median of all replicate wells ($Diff_{well}^{robust\,z\text{-}score}$) were calculated according to Eq. 4.

$$Diff_{well}^{robust\,z\text{-}score} = X_{single\,well}^{robust\,z\text{-}score} - Median_{AllRep\,wells}^{robust\,z\text{-}score} \quad (4)$$

$X_{single\,well}^{robust\,z\text{-}score}$ = Well-level robust z-score of a given feature from a single well;

$Median_{AllRep\,wells}^{robust\,z\text{-}score}$ = Median of the well-level robust z-scores of a given feature from AllRep wells ($N_{Tech} = 3$ and $N_{Biol} = 4$)

To visualize the intra-/inter-plate variabilities for each feature and replicate experiment, the differences of the well-level robust z-scores (from Eq. 4) were aggregated to the replicate-level by calculating for each feature the median of well-level differences ($MedianDiff_{replicate}^{robust\,z\text{-}score}$, equation 5) for each technical (TRep), biological (BRep), or all (AllRep) replicate experiment. Those median replicate-level differences (from *equation 5*) were plotted as heatmaps to generate compound activity profiles showing the intra-/inter-plate variabilities of all features per compound and concentration for the technical (TRep) or biological (BRep) replicate experiments conducted for CPP and CP at 20x and 40x magnification.

To quantify the total intra-/inter-plate variabilities of all features for each replicate experiment, the absolute values of the differences of the well-level robust z-scores (from Eq. 4) were first summed up for each well and then aggregated to the replicate-level to calculate the sum replicate-level differences ($SumDiff_{replicate}$) for each technical (TRep), biological (BRep), or all (AllRep) replicate experiment according to Eq. 6.

$$SumDiff_{replicate} = \sum_{well\,robust\,z\text{-}score}^{n} \sum^{m} |Diff_{well}^{robust\,z\text{-}score}| \quad (6)$$

robust z-score = single feature; m = number of features
well = single replicate well; n = number of replicate wells;
$|Diff_{well}^{robust\,z\text{-}score}|$ = Absolute value of the difference of the well-level robust z-score of a given feature from a single (technical or biological) replicate well

Since the calculated sum of replicate-level differences was directly related to the number of replicate experiments conducted and the number and effect sizes of features included in CPP and CP, normalization to the sum of replicate-level responses was required. Therefore, the absolute values of the well-level robust z-scores (from Eqs. 2 and 3) were first summed up for each well and then aggregated to the replicate-level (analogous to Eq. 6) to calculate the sum replicate-level responses ($SumResp_{replicate}$) for each technical (TRep), biological

(BRep), or all (AllRep) replicate experiment according to Eq. 7.

$$SumResp_{replicate} = \sum_{well\ robust\ z\text{-}score}^{n} \sum^{m} |X_{well}^{robust\ z\text{-}score}| \qquad (7)$$

robust z-score = single feature; m = number of features
well = single replicate well; n = number of replicate wells;
$|X_{well}^{robust\ z\text{-}score}|$ = Absolute value of the well-level robust z-score of a given feature from a single (technical or biological) replicate well

The sum replicate-level responses were finally used for normalization of the sum replicate-level differences to calculate the relative sum of replicate-level differences ($RelSumDiff_{replicate}$) according to Eq. 8.

$$RelSumDiff_{replicate} = \frac{SumDiff_{replicate}}{SumResp_{replicate}} \qquad (8)$$

$SumDiff_{replicate}$ = Sum replicate-level differences of the well-level robust z-scores of all features from <u>all</u> related (technical or biological or all) replicate wells (3 or 4 or 12 replicate wells per screen)
$SumResp_{replicate}$ = Sum replicate-level differences of the well-level robust z-scores of all features from <u>all</u> related (technical or biological or all) replicate wells (3 or 4 or 12 replicate wells per screen)

As a measure for the total intra-/inter-plate variabilities, those relative sum replicate-level differences (from Eq. 8) were finally reported in table format for the technical (TRep), biological (BRep), or all (AllRep) replicate experiments conducted for CPP and CP at 20x and 40x magnification.

**Benchmark concentration (BMC) modeling.** For each feature, the well-level robust z-scores (from *equation 2* and Eq. 3) were also used for benchmark concentration (BMC) modeling using the curve-fitting R package *tpclfit2*[80] as previously described[8,12]. BMCs were only determined from at least four non-cytotoxic concentrations of at least three biological replicates to ensure high quality BMC curve fitting in accordance with published guidance on benchmark dose modeling in toxicology[81]. Curve-fitting was performed on four biological replicates (median of well-level robust z-scores from each three technical replicates) using five different curve-fitting models (cnst, hill, poly1, poly2, pow). The best-fitting curve was selected to calculate the BMC of each individual feature, which corresponded to the concentration at which the phenotypic response exceeded the defined benchmark response (BMR) cutoff. The BMR was specifically calculated for each feature from the corresponding well-level MAD of the DMSO solvent control and was set at $BMR = 1.349*SD$ (corresponding to approximately 1 interquartile range (IQR) for normal distributed data) as described before[48], with $SD = 1.4826*MAD_{ctrl\ wells}^{feature}$. Features for which no BMC could be determined (BMC$^{NA}$) or with estimated BMCs greater than the highest tested non-cytotoxic concentration (BMC$^{HIGH}$) were considered inactive. Conversely, in cases where the estimated BMC was smaller than the lowest tested concentration (BMC$^{LOW}$), the BMC was adjusted to a modified lowest concentration (half-log smaller than the lowest tested concentration) according to Eq. 9.

$$\log 10(BMC^{LOW}) = \log 10(lowest\ tested\ conc.) - 0.5 \qquad (9)$$

To support BMC data analysis, each of the CPP/Harmony, CP/Harmony, CPP/Cell Profiler features were assigned to 62 (CPP/Harmony), 41 (CP/ Harmony), or 68 (CPP/Cell Profiler) biologically meaningful feature categories, respectively, representing specific combinations of channels, cell regions, and analysis modules (Supplementary Data 5) in a similar way as previously described for CP[8,11,12,17].

These BMC data were visualized using Proportion BMC profiles (category-level), accumulation plots (category-level), and magnitude plots (feature-level). The Proportion BMC profiles were plotted using the R software [v4.2.2] and display the proportion of single features per feature category showing significant ($p < 0.05$, one-sided Student's t-test) responses (log10 BMC) to treatment with a particular reference compound compared to all tested compounds. For generation of Proportion BMC profiles, features for which no BMC could be determined (BMC$^{NA}$) were adjusted to a modified BMC corresponding to the maximal tested compound concentration (100 μM) to enable comparison between all compounds. Accumulation and magnitude plots were generated as previously described for CP[8,11,12,17]. Each data point in the accumulation plot is a different feature category represented by a ranked median category BMC (median of all feature BMCs constituting a feature category, with at least 30% of features showing a response to compound treatment). In the corresponding magnitude plot, each data point is a different feature represented by the BMC and the normalized magnitude (i.e., the maximum robust z-score in the tested concentration range).

### Data visualization and statistical analyses
All quantitative data were exported into Excel (Microsoft, Redmond, WA, USA)-readable (.*csv*) files. Harmony analysis sequences were exported into web archive (.*mht*) files. Compound activity profiles, BMC accumulation plots, and BMC magnitude plots were exported into vector graphic (.*svg*) files. Proportion BMC plots were exported into portable document format (.*pdf*) files. Quantitative data were plotted using Prism 10 (GraphPad Software, San Diego, CA, USA) or using the R software [v4.2.2]. Statistical analyses (Pearson/Spearman correlation, hierarchical clustering) of robust z-score and BMC (log10 BMC) data were performed using the R software [v4.2.2]. The individual R scripts are available as Supplementary Data 22 and publicly available on Zenodo (https://doi.org/10.5281/zenodo.14982928). Figures were generated using Illustrator CC 2024 (Adobe, San Jose, CA, USA).

### Reporting summary
Further information on research design is available in the Nature Portfolio Reporting Summary linked to this article.

## Data availability
The processed data (aggregated and processed profiles and BMC data) generated in this study are provided in the Supplementary Data/ Source Data files and are publicly available in the Zenodo repository under accession code 14982928[82]. Due to their large size (25-50 GB per plate), the raw data (images and unprocessed profiles at the single-cell level) are available upon request directed to the corresponding authors.

## Code availability
The image analysis pipelines (Harmony and Cell Profiler software) as well as KNIME pipelines and R code for data analysis and generation of plots for figures in this study are provided in the Supplementary Data file and are publicly available in the Zenodo repository under accession code 14982928[82]. The individual Harmony sequences for CP and CPP image analysis are available as Supplementary Data 19. The individual Cell Profiler sequences for CPP image analysis, e.g., for illumination correction and image analysis, are available as Supplementary Data 20. The individual KNIME pipelines, e.g., for data standardization, generation of compound activity profiles, evaluation of phenotypic profile variabilities, BMC modeling, generation of accumulation and magnitude plots, are available as Supplementary Data 21. The individual R scripts are available as Supplementary Data 22. The *4i stitcher* software [v1] is publicly available in the Zenodo repository under accession code 13784742[83]. The *Add Channels 4i* ABB [v1.0.4] is publicly available in the Zenodo repository under accession code 15119993[84].

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

## Acknowledgements

We extend our gratitude to Joshua A. Harrill and Jo Nyffeler for generously providing a CP Harmony analysis pipeline and R code for data analysis, and for sharing their insights and experiences with CP. Special thanks to Achim Kirsch (Revvity Inc., Waltham, MA, USA) for his invaluable support in the development of the Harmony application-specific building block (ABB) for CPP, and Bob van de Water (Leiden University, The Netherlands) for supplying the RPTEC-TERT1 cell line. We also acknowledge our colleagues from BfR/Bf3R for their scientific input and valuable comments on the manuscript. Intramural funding of the German Federal Ministry of Education and Research (BMBF) was provided to S.D. for the MORPHEUS Project (Grant No. 16LW0137K). Intramural

funding of the European Union's Horizon 2020 Research and Innovation Programme was provided to S.L. for the RISK-HUNT3R Project (Grant No. 964537), which is part of the ASPIS cluster.

## Author contributions

S.L. and S.D. conceived the project and devised the research plan. E.C., M.W. performed the experiments. J.M.M., S.L., S.D., E.C. and M.W. conducted image and data analysis. N.K. established the image registration *4i stitcher* software. C.W. generated compound plates and supported image and data analysis. E.C., S.L., and S.D. wrote the main manuscript text, and prepared figures and tables. All authors reviewed the manuscript.

## Funding

## Competing interests

The authors declare no competing interests. The German Federal Institute for Risk Assessment (BfR) is a scientifically independent institution within the portfolio of the Federal Ministry of Food and Agriculture (BMEL) in Germany. The authors' freedom to design, conduct, interpret, and publish research is explicitly not compromised. The opinions expressed in this document reflect only the author's view. The European Commission is not responsible for any use that may be made of the information it contains.
