## [Transparent Peer Review file · Nature Communications]

Cell Painting PLUS: Expanding the multiplexing capacity of Cell Painting-based phenotypic profiling using iterative staining-elution cycles

Corresponding Author: Dr Sebastian Dunst

Version 0:

Reviewer comments:

Reviewer #1

(Remarks to the Author)

The authors have developed an assay similar to Cell Painting (CP), called Cell Painting PLUS (CPP), and have demonstrated its comparability with and advantages over Cell Painting. The CPP assay uses 7 dyes and images 9 compartments of a cell in separate imaging channels. The cells are imaged in 2 imaging cycles by eluting out the first set of dyes using an optimal elution buffer. The advantage of imaging the compartments in separate channels is well demonstrated with the profiles of tetrandrine and fluphenazine. Overall, the CPP assay has the advantages of multiplexing capacity, versatility, and specificity obtained due to imaging in individual channels. While ultimately valuable, the similarity to previous work and the results that show similar performance to the existing CP assay may mean this work does not clear the bar to "advance understanding in a way that will move the field forward". Nevertheless, the manuscript is nonetheless of excellent technical quality and we can provide only minor suggestions (as mentioned below) for better clarity to the readers.

- The authors mention that manual verification was needed for image registration when the cross-correlation coefficient went below 0.8 in the CellProfiler pipeline. Was such a step needed in Harmony as well? Or did the ABB that was used in Harmony to correct image registration have a way to handle such images?

- More details on the Python software 4i stitcher should be made available such that the methodology is possible to adapt. Can the authors comment on whether registration within the CellProfiler pipeline was attempted, removing the need for an extra piece of software, and with what result?

- Including the KNIME pipelines as supplementary information will make it easier for the readers to reproduce the methodology.

- Can authors comment on how not performing feature reduction or feature selection increases transferability?

Reviewer #2

(Remarks to the Author)

"I co-reviewed this manuscript with one of the reviewers who provided the listed reports. This is part of the Nature Communications initiative to facilitate training in peer review and to provide appropriate recognition for Early Career Researchers who co-review manuscripts."

Reviewer #3

(Remarks to the Author)

In this report, the authors propose modifications in the Cell Painting (CP) method, which they claim expands the multiplexing capacity of CP-based phenotypic profiling. They do so by applying the stains in sequence, with an elution step in between (dubbing the new method CP Plus, or CPP). This allows them to add a stain for lysosomes, and to improve spectral and channel separation between the various fluorescent dyes. The assay development work is commendably rigorous and thorough. Assay design is also rigorous and well-controlled.

That said, the manuscript strains itself to find a significant edge for CPP over CP, but that proves hard to come by, particularly when one puts this in the context of high throughput screening (the manuscript purports this to be a better way for high throughput phenotypic profiling (HTPP)). If there is indeed an edge for CPP over CP, it appears to be more on the side of MoA dissection. However, HTPP, like all HTS, is essentially an optimization in trade-offs. For large-scale screening, CPP injects a substantial additional cost for reagents, time, and instrument times (liquid handling, imaging, etc). This additional cost in time and resources is hardly warranted by whatever additional MoA insights CPP seems to be contributing. MoA dissections are much better suited for follow-up characterizations of hits from a primary screen. Making the primary screen prohibitively expensive, for added MoA insights, is not the modus operandi of experienced screening entities. One of the main reasons CP is a desirable approach is because it has a high information density to labor/cost ratio. Doubling the labor and cost for a minor improvement in MoA insight greatly undercuts this mantra and is not likely to be adopted by HTPP. If the authors are intent on framing this as an HTPP method that can outperform CP, then they might benefit from backing that up with numbers. What surcharge of labor/instrument time/reagents will come from performing an actual HTPP (thousands or millions of compounds, not just 30) with CPP instead of CP? Is that cost worth the MoA insights that one can glean from CPP? Or does it make more sense to run HTPP with CP, and then perform CPP as a secondary assay with hit compounds?

Specific critiques:

1- QUOTE: "116 All CPP dyes were well detectable over the course of 4 weeks after staining, with some fluctuations in the staining intensities observed for the Lyso and ER dyes from day 2 after staining (Supplementary Fig. 1B)."

A fluctuation is a change between alternating states. As such, the temporal change observed for the ER stain cannot be handwaved as a "fluctuation". Rather, it appears to be a sustained and stable increase. If this is a real observation, it needs to be better explained/discussed.

2- QUOTE: "186 The image analysis procedure included the registration and segmentation of images (Supplementary Fig. 2A), followed by the feature extraction (Supplementary 188 table 3, 5) using either the commercial Harmony software (Revvity Inc.) (894 features) or the open-source Cell Profiler software (3,044 features)."

The fact that the choice of feature extraction software is not made explicit throughout the manuscript makes it ambiguous at times. Also, since the authors went through the trouble of using both feature extraction methods, it would be good to compare the performance of the two. As someone who has access to both, this reviewer would for one appreciate an analysis demonstrating whether one would be superior to the other. If they lend themselves equally well to the methodology, that would be good to know as well.

3- QUOTE: "209 profiles for the DNA and Mito channels (Supplementary Fig. 2C). However, the activity profiles of fluphenazine and tetrandrine indicated clear Mito activities in CP but not 211 CPP (Fig. 2C-D, Supplementary Fig. 2C). Although those Mito activities were consistent with published CP profiles of MCF-7 cells, they may not represent the most relevant activity when considering the reported, primarily Lyso-related MoA for the two compounds (Fig. 2A)."

This claim seems unsubstantiated. Just because the compounds have lyso- activities reported doesn't necessarily mean that they do not have mito- activity. Rather than attempt to explain why CP mito captures an event and CPP mito doesn't, the authors have opted to discount the possibility that these compounds have mito- effects. This warrants further investigation to understand what the ground truth here is (is there a mito- effect or not?), and then decide based on that which method is capturing the ground truth better.

4- QUOTE: "173 Barcoding compound effects using CPP improves the specificity of phenotypic profiles"

This claim appears to be unsubstantiated by the data and mostly bolstered by the use of compounds with lyso-related MoA. It is not clear from the current data that CPP would be better at large-scale comparisons with compounds of all MoAs.

5- QUOTE: "248 increased data variability to some extent. To facilitate analysis of the BMC modeling data, we further assigned each of the 894 (CPP) or 558 (CP) features to 62 (CPP) or (CP) biologically meaningful feature categories representing specific combinations of channels, cell regions, and analysis modules (Supplementary table 5; Supplementary table 6) in a similar way as previously described for CP. "

The 894 and 558 are extracted how? Harmony? Or CellProfiler?

6- QUOTE: "281 To further distinguish between early (most sensitive, potentially primary) and later (less sensitive, potentially secondary) compound effects, we further investigated the specific sequence of feature category responses as well as the maximum sizes of those effects for CPP (Fig. 3D; Supplementary Fig. 4A-O) and CP (Supplementary Fig. 5A-O) using accumulation and magnitude plots as previously described for CP."

This is confusing because it refers to temporal effects (early vs late) but instead, the actual analysis is concentration, not time-based. Perhaps the authors are referring to "specific/low concentration" vs "non-specific/high concentration" effects here? Either way, calling it early/late might be confusing to readers.

7- QUOTE: "294 with its broad activity shown in the Proportion BMC profile (Fig. 3B). In particular, CPP was able to

distinguish between RNA and ER features showing maximum effect sizes, highlighting the advantage of separating RNA and ER channels to more precisely determine different MoA.”

When RNA and ER are separated by cell compartment (not just by channel), CP appears to capture distinct changes in RNA and ER as well (Fig 3B “Channels + Regions”).

8- QUOTE: “Rapamycin, a highly specific mTOR inhibitor, caused a strong effect on the number of nucleoli per 317 cell (Fig. 4A; Supplementary Fig. 4I), consistent with the role of the mTOR pathway in controlling cell cycle progression through regulating ribosome biogenesis and nucleoli numbers.”

The cellular IC50 for mTOR is in the low nanomolar range. The tested concentration of 10 uM is too high, and more so for a 48 hours assay. It is known that prolonged treatment with Rapamycin (after 24 hours) has inhibitory effects on mTORC2 activity.

9- QUOTE: “412 However, when using the CP method, these compounds exhibited pronounced responses in the combined RNA/ER channel (Supplementary Fig. 5E, L, O), indicating that the morphological changes in the RNA/ER channel in the CP method mainly reflect those in the ER channel in the CPP assay, highlighting the advantage of separating RNA and ER channels to more precisely determine different MoA”

This is stating the obvious – a channel that combines stains for two components will show the combined effects on both components. The question should be whether adding compartment analysis to this channel (nucleus vs cytoplasm) in CP would distinguish between effects on RNA vs effects on ER. In this case, this advantage is not entirely lacking from CP.

10- QUOTE: “423 In this proof-of-concept study, we established an efficient, robust, and broadly applicable HTPP assay for multiplexed, iterative staining of single cells with at least seven fluorescent dyes labeling at least nine different subcellular compartments and organelles.”

Per previous comments above, this review does not agree with this. If anything, the method was de-optimized for HTPP. Rather, it was optimized for having more explanatory and mechanistic power than CP (maybe), but the trade-off against HTPP is quite large.

11- QUOTE: “489 It is further important to consider that, in comparison to CP, the CPP assay necessitates (at least) one additional elution, staining, and imaging cycle, thereby demanding larger data storage and high-performance computing capacity. However, working with cells at higher confluency would reduce the number of imaging fields and thus also the required amount of imaging data.”

Imaging fewer fields does not counteract the added burden of liquid handling and reagent costs. It might trade off statistical robustness.

12- The creation of CPP from CP is enabled by the elution buffer, which the authors claim is superior to other elution buffers for this application. Thus, the development of this buffer is a central technological component of this method. However, details about this buffer, its composition, and the process by which it was selected and optimized are completely lacking in the manuscript.

13- The authors attempt to give the manuscript a mechanistic/discovery angle, but the observations are often shallow and unsubstantiated by the data presented. In-depth mechanistic investigations are lacking for observations to be meaningful.

Version 1:

Reviewer comments:

Reviewer #1

(Remarks to the Author)

We thank the author for their response to our initial criticisms. However, their responses about software are somewhat troubling - not performing a feature reduction step on CellProfiler features is not the accepted way to perform the assay (Bray 2016, Caicedo 2017, Cimini 2023, in addition to the authors own citations of Chandrasekaran, Rohban, and Way), so any comparisons between CellProfiler and Harmony become immediately suspect when the CellProfiler feature set is not used per long-established best practices. Feature reduction is a very simple procedure which can, if maximum comparability across two data sets taken in two locations is desired, be performed in a single line of code by simply importing the column list from one data set to the other. Such a simple transition is notably NOT possible if one location has Harmony and one does not because it uses another brand of microscope, since most if not all high-content imagers can successfully perform Cell Painting (Bray 2016, Cimini 2023, Tromans-Coia 2023).

We are otherwise convinced generally convinced by the arguments of reviewer 2 who argues convincingly that doubling the reagent costs of a primary screen (per the authors own admission) is unlikely to gain wide adoption in a context where the one of the largest benefits of Cell Painting is its low cost and therefore high scalability. We therefore find this manuscript of

sufficient technical quality but questionable significant addition to the field.

Reviewer #2

(Remarks to the Author)

Reviewer #3

(Remarks to the Author)

Version 2:

Reviewer comments:

Reviewer #1

(Remarks to the Author)

We thank the authors for adding the figures where CellProfiler features have undergone feature reduction, which allows a more uniform comparison with Harmony features. However, as per previous rounds of review we still disagree with the notion that Harmony features are "more consistent comparisons between different screens and laboratories" since Harmony is an expensive commercial software which only comes on certain brands of microscope, and CellProfiler features can and have been used on Cell Painting data from many brands of microscope, and one can compare data across sites by simply copying another group's final feature list in a single line of Python (or R, etc) code. This is the last though of our technical concerns.

Reviewer #2

(Remarks to the Author)

Response to reviewer comments on von Coburg et al.

“Cell Painting PLUS: Expanding the multiplexing capacity of Cell Painting-based phenotypic profiling using iterative staining-elution cycles”

Comments received by authors: 08 Aug 2024

Response submitted by authors: 15 Nov 2024

We are very grateful for the very constructive feedback from the reviewers and we have revised the manuscript according to their suggestions.

During the constant process of improving our KNIME workflows, we thoroughly double check, simplify, and annotate the processing steps in each of the pipelines to be readily applicable for other users. During this process, we noticed that, in our initially submitted manuscript, the robust z-score calculation (data normalization) was done incorrectly. From the node description in KNIME, it was initially not clear to us that the scaling factor 1.4826 (see methods section) was already embedded in the “Normalize Plates (Z-Score)” KNIME node¹ that we used. Because we manually added the scaling factor to the robust Z-score calculation, we ended up using it twice instead of just once. This is corrected now. The data have been re-analyzed with the corrected pipeline and figures and other results were updated. During this correction process, we used the chance to also improve the visualization of the three “Nucleoli Number” features (i.e., “1-2”, “3-4”, “>5”) described in Fig. 4 of the manuscript by calculating the number of cells per “Nucleoli Number” feature relative to the total number of all cells. None of the conclusions drawn from the initially submitted manuscript were affected by these corrections.

Meanwhile, our KNIME workflows, including all nodes provided by others, have been reviewed in detail by four scientists to ensure high quality of our workflows, which we release together with this publication.

¹ <https://hub.knime.com/mpicbg-tds/extensions/de.mpicbg.tds.knime.hcstools.feature/latest/de.mpicbg.knime.hcs.base.nodes.norm.zscore.ZScoreNormalizerNodeFactory>

<https://github.com/knime-mpicbg/HCS-Tools/blob/dd07ec2638c4650bdc032cc7458b13d8d7b2f02b/de.mpicbg.knime.hcs.base/src/de/mpicbg/knime/hcs/base/nodes/norm/zscore/ZScoreNormalizerNodeModel.java>

Reviewer #1:

General comments:

The authors have developed an assay similar to Cell Painting (CP), called Cell Painting PLUS (CPP), and have demonstrated its comparability with and advantages over Cell Painting. The CPP assay uses 7 dyes and images 9 compartments of a cell in separate imaging channels. The cells are imaged in 2 imaging cycles by eluting out the first set of dyes using an optimal elution buffer. The advantage of imaging the compartments in separate channels is well demonstrated with the profiles of tetrandrine and fluphenazine. Overall, the CPP assay has the advantages of multiplexing capacity, versatility, and specificity obtained due to imaging in individual channels.

While ultimately valuable, the similarity to previous work and the results that show similar performance to the existing assay may mean this work does not clear the bar to "advance understanding in a way that will move the field forward". Nevertheless, the manuscript is nonetheless of excellent technical quality and we can provide only minor suggestions (as mentioned below) for better clarity to the readers.

Answer:

We agree with the reviewer's opinion that the CPP assay is based on the CP assay and is not an entirely new method. However, we think that the complete separation of imaging channels and inclusion of lysosome staining (enabled by the development of the dye elution buffer) does provide a basis for more precise and specific phenotypic profiling based on iterative staining-elution cycles.

For example, in our case studies on cytochalasin D/latrunculin B (Use case 2; shown in Fig. 5, Fig. S6) and sunitinib malate/siramesine/tetrandrine/fluphenazine (Use case 3; shown in Fig. 6, Fig. S7-10) we demonstrate the capability of CPP to identify distinct alterations in morphological Actin and Golgi vs AGP and ER vs RNA/ER features, which cannot be distinguished in the same cell regions by CP. In the new revised version, we further improved the respective sections by adding more in-depth analyses of our data to further clarify the advantages of CPP (see our response to specific comment 13 of reviewer 3).

Specific points:

1.) *The authors mention that manual verification was needed for image registration when the cross-correlation coefficient went below 0.8 in the Cell Profiler pipeline. Was such a step needed in Harmony as well? Or did the ABB that was used in Harmony to correct image registration have a way to handle such images?*

Answer:

When using Harmony, there was no need for this manual verification. To elaborate on this aspect, we added to the SI:

- **Image analysis and feature extraction/A) Harmony: *“The correction model that was used in Add Channels 4i has a narrow acceptance range for parameters. A maximum shift of 50 pixel in x and y direction (< 5% of image width) is assumed, and no scaling nor rotations are expected. In performance tests, the correlation was always sufficiently good that no manual check was deemed necessary.”***

2.) *More details on the Python software 4i stitcher should be made available such that the methodology is possible to adapt. Can the authors comment on whether registration within the Cell Profiler pipeline was attempted, removing the need for an extra piece of software, and with what result?*

Answer:

We agree with the reviewer. The main reason for using custom code compared to Cell Profiler modules is the greater flexibility. For example, Cell Profiler has an "align" module that also uses the cross-correlation coefficient, as we did. However, we also added optional rotational registration for small angles to account for very slight rotational shifts that may occur between runs. In contrast, Cell Profiler only allows translational shifts. In addition, the cross-correlation coefficient is not provided by Cell Profiler but by the 4i stitcher. This cross-correlation coefficient can be used as measure for the quality of the registration. The detailed description of the 4i stitcher method was included in the SI under the subheading “B) Cell Profiler”. In the revised SI, we added the following information:

- Image analysis and feature extraction/B) Cell Profiler: *“For future development, this python-based code can be easily adopted for different microscope systems and/or extended for more advanced (pre-)processing steps. To enable adoption of the methodology, the 4i stitcher software is publicly available on Zenodo (<https://zenodo.org/records/13784742>).”*

3.) Including the KNIME pipelines as supplementary information will make it easier for the readers to reproduce the methodology.

Answer:

Thank you for this advice. With this manuscript, we release all Harmony and Cell Profiler image analysis sequences, the KNIME pipelines, and the R scripts used. We added the following text to the SI:

- Image analysis and feature extraction/A) Harmony: *“The individual Harmony sequences for CP and CPP image analysis are available as Supplementary Material 14.”*
- Image analysis and feature extraction/B) Cell Profiler: *“The individual Cell Profiler sequences, e.g., for illumination correction and image analysis, are available as Supplementary Material 15.”*
- Data analysis: *“The individual KNIME pipelines, e.g., for data standardization, generation of compound activity profiles, evaluation of phenotypic profile variabilities, BMC modeling, generation of accumulation and magnitude plots, are available as Supplementary Material 16.”*
- Data visualization and statistical analyses: *“The individual R scripts are available as Supplementary Material 17.”*

In addition, we plan to upload the KNIME workflows to Zenodo and will add the links to the SI when the article is accepted for publication.

4.) Can authors comment on how not performing feature reduction or feature selection increases transferability?

Answer:

This is a very relevant question. We intentionally did not perform feature reduction in this study. The primary aim of this paper is to introduce the CPP methodology and to compare it with CP in order to explore its respective added value for specific research questions. Our ongoing projects aim to further compare CPP results across different laboratories, cell lines, and cultivation conditions. In our experience, feature reduction based on correlation often yields different reduced feature sets across different laboratories and even across replicates. As a result, evaluating reproducibility across biological/technical replicates and comparing results from different screening sites becomes challenging when different features are removed through feature reduction. Additionally, maintaining the same data architecture across screenings avoids the need for adapting downstream automated data analysis procedures such as phenotypic profile clustering and BMC modeling.

When establishing the Harmony and Cell Profiler image analysis pipelines for extracting cellular features from CPP and CP images, we only selected relevant combinations of channels, regions, and modules (CP/Harmony: 558 features, CPP/Harmony: 894 features, CPP/Cell Profiler: 3,648 features). In the data cleaning process during data normalization, no Harmony features with overall constant or missing values were excluded from the dataset (in contrast to our Cell Profiler pipeline), supporting the relevance of the selected Harmony features and avoiding the need for performing feature reduction.

In the revised version of the manuscript, we address this point as follows:

- Discussion: *“Moreover, from the large number of features being extracted when using Cell Profiler, ~600 features had to be filtered out because of a high number of missing or identical values. This made the use of Cell Profiler less convenient, as it required more extensive initial filtering of features and may consequently produce different feature lists for individual experiments. Previous reports from several laboratories have also noted the need to filter and reduce features when using Cell Profiler [Chandrasekaran, 2024; Rohban, 2017; Way, 2022], and with our analysis we concur with these findings. In turn, the selected features from Harmony proved reliable for downstream analysis without the need for feature reduction or selection, facilitating more consistent comparisons across different screens and laboratories.”*

Reviewer #2:

General comments:

co-reviewed this manuscript with one of the reviewers who provided the listed reports. This is part of the Nature Communications initiative to facilitate training in peer review and to provide appropriate recognition for Early Career Researchers who co-review manuscripts.

Answer:

Thank you very much for your valuable feedback on our manuscript.

Reviewer #3:

General comments:

In this report, the authors propose modifications in the Cell Painting (CP) method, which they claim expands the multiplexing capacity of CP-based phenotypic profiling. They do so by applying the stains in sequence, with an elution step in between (dubbing the new method CP Plus, or CPP). This allows them to add a stain for lysosomes, and to improve spectral and channel separation between the various fluorescent dyes. The assay development work is commendably rigorous and thorough. Assay design is also rigorous and well-controlled.

Answer:

Thank you very much for your supporting feedback regarding the rigor of our work.

That said, the manuscript strains itself to find a significant edge for CPP over CP, but that proves hard to come by, particularly when one puts this in the context of high throughput screening (the manuscript purports this to be a better way for high throughput phenotypic profiling (HTPP)). If there is indeed an edge for CPP over CP, it appears to be more on the side of MoA dissection. However, HTPP, like all HTS, is essentially an optimization in trade-offs. For large-scale screening, CPP injects a substantial additional cost for reagents, time, and instrument times (liquid handling, imaging, etc). This additional cost in time and resources is

hardly warranted by whatever additional MoA insights CPP seems to be contributing. MoA dissections are much better suited for follow-up characterizations of hits from a primary screen. Making the primary screen prohibitively expensive, for added MoA insights, is not the modus operandi of experienced screening entities. One of the main reasons CP is a desirable approach IS because it has a high information density to labor/cost ratio. Doubling the labor and cost for a minor improvement in MoA insight greatly undercuts this mantra and is not likely to be adopted by HTPP. If the authors are intent on framing this as an HTPP method that can outperform CP, then they might benefit from backing that up with numbers. What surcharge of labor/instrument time/reagents will come from performing an actual HTPP (thousands or millions of compounds, not just 30) with CPP instead of CP? Is that cost worth the MoA insights that one can glean from CPP? Or does it make more sense to run HTPP with CP, and then perform CPP as a secondary assay with hit compounds??

Answer:

Thank you very much for your valuable suggestions to elaborate on the positioning of the novel CPP method in the landscape of the available HTPP methods. We would like to emphasize that we consider CPP as a valuable addition and alternative approach to CP but not as a replacement. With the development of CPP, we aimed at expanding the repertoire of available HTPP methods in order to offer researchers a customizable assay that provides higher profile diversity and, thus, enhanced MoA information, if needed to address the specific research question. We do agree with the reviewer that not every research question might require the separation of the AGP and RNA/ER channels or addition of the Lyso channel, and thus may not justify additional investments of cost, time, or resources. But for other research questions, the benefit to investigate these channels independently could overcome the additional investment involved. Therefore, depending on the scientific question, scientists need to be able to choose the best suitable assay. To make our intended positioning of CPP clearer, we have addressed and revised the manuscript at multiple instances. The revised main statements are given here:

- ***Abstract: “Here we present the Cell Painting PLUS (CPP) assay, an efficient, robust and broadly applicable approach that further expands the versatility of available HTPP methods and offers additional options for addressing mode-of-action specific research questions.”***

- **Abstract:** *“In this way, CPP significantly expands the flexibility, customizability, and multiplexing capacity of the original CP method and, importantly, also improves the organelle-specificity and diversity of the phenotypic profiles due to the separate imaging and analysis of single dyes in individual channels.”*
- **Introduction:** *“To maximize HTPP capacity while maintaining a very high information density, signals from two CP dyes are often intentionally merged in the same imaging channel (i.e., RNA+ER and/or Actin+Golgi), accepting the trade-off that this optimization may compromise the organelle-specificity of the phenotypic profiles. To add a new approach to the versatile HTPP methods, which offers additional options for addressing more MoA-specific research questions, we therefore developed the Cell Painting PLUS (CPP) assay. CPP significantly expands the flexibility, customizability, and multiplexing capacity of the original CP method, and, importantly, also improves the organelle-specificity and diversity of the phenotypic profiles.”*
- **Results/Subheading 3:** *“Barcoding compound effects using CPP expands the diversity of phenotypic profiles”*
- **Results:** *“In conclusion, this case study suggests that separating Actin and Golgi channels can be beneficial to enhance the organelle-specificity and diversity of phenotypic profiles for comparative MoA analyses. The example of cytochalasin D and latrunculin B highlights the need for generating phenotypic profiling data across a concentration range and using different analysis methods for detailed MoA analyses.”*
- **Discussion:** *“The CPP method thus adds a new approach to the versatile HTPP methods and offers additional options for addressing MoA-specific research questions.”*
- **Discussion:** *“The case studies indeed suggested that separating Actin and Golgi as well as ER and RNA channels and, in particular, addition of the Lyso channels may be beneficial to enhance the organelle-specificity and diversity of phenotypic profiles for comparative MoA analyses. However, the number of compounds for different MoA provided in this study was yet limited and the first indications regarding a potential higher phenotypic profile-specificity of CPP over CP will need to be strengthened by*

comparing the reference compound profiles against a larger set of compounds with known MoA.”

- Discussion: *“With developing the CPP method, we aimed to provide the growing Cell Painting community with a more customizable version of the very successful CP approach with increased organelle-specificity of the phenotypic profiles, accepting the trade-off that this optimization reduces the high-throughput capacity of CPP compared to CP. Since not every research question may require the separation of the AGP and RNA/ER channel or addition of the Lyso channel, it will be up to the individual scientist to decide, if the advantages of CPP serve their specific research needs and outweigh a higher investment in cost, time, and resources compared to CP.”* (this part is also covering the aspect on cost/labor discussed next)

We would also like to highlight that, in our hands, the additional experimental labor for the CPP assay is about 1h for preparation of dye/elution buffer working solutions, the elution procedure (15 min), and cycle 2 staining (35 min). Of course, the total additional time required for CPP depends heavily on site- and screen-specific factors, such as the number of plates to be tested, the level of automation of the liquid handling and imaging routines, and the individual imaging settings. Regarding reagent costs, apart from the dyes commonly used in both CP (~54€/384 well plate) and CPP (~56€/384 well plate) at comparable concentrations, CPP adds expenses for the additional Lyso dye (50€/384 well plate) and only minor costs for elution (€1/384 well plate). Overall, this adds up to reagent costs of 107€ for a 384-well plate screened with CPP or 54€ screened with CP (excluding the fix costs for robotic pipetting tips, well plates etc.). In the future, it may be possible to exchange the Lyso dye with a cheaper alternative, however, we so far could not find a suitable replacement of that dye that was compatible with fixed cell staining. To make this aspect on experimental labor and reagent costs clearer to the users of the CPP method, we have now summarized and compared the experimental labor and exact reagent costs for CPP and CP in Supplementary Table 1, and updated the manuscript as follows:

- Results: *“The dye concentrations used in CPP and in the original or recently updated CP method were similar indicating comparable screening costs in CPP and CP per single dye used (Supplementary table 1). Thus, the additional reagent costs of the*

CPP assay are mainly due to the inclusion of the Lyso dye (Supplementary table 1), but may decrease if alternative Lyso dyes compatible with fixed cell staining become available in the future.”

- Results: *“Thus, the CPP elution buffer provided not only an efficient, but also a time- and cost-effective elution procedure in the CPP protocol (Supplementary Table 1).”*
- Discussion: *“With developing the CPP method, we aimed to provide the growing Cell Painting community with a more customizable version of the very successful CP approach with increased organelle-specificity of the phenotypic profiles, accepting the trade-off that this optimization reduces the high-throughput capacity of CPP compared to CP. Since not every research question may require the separation of the AGP and RNA/ER channel or addition of the Lyso channel, it will be up to the individual scientist to decide, if the advantages of CPP serve their specific research needs and outweigh a higher investment in cost, time, and resources compared to CP.”*

Moreover, the here described screening was performed without any prior cytotoxicity pre-screening of the 15 tested compounds. Instead, compounds were screened at eight concentrations each, covering both cytotoxic and non-cytotoxic concentrations. This was intended to allow comparability to previous CP-based publications (e.g., Willis et al., 2020), to explore the method’s potential for either primary or secondary screening, and to identify potential parameters that can help optimizing the balance between cost, time and resources. For future large-scale screenings, this overall comprehensive experimental design should be complemented by a prior well-established, easy-to-perform, and cost-effective cytotoxicity screening using, e.g., Propidium Iodide (PI), MTT, or Alamar Blue assays, in order to significantly reduce the number of relevant testing conditions. The in this study used experimental design of testing multiple concentrations per compound and in multiple technical/biological replicates enabled the comparison of compound activity profiles at one maximum non-cytotoxic concentration (see our response to the specific comment 13 from reviewer 3), but also the extraction of concentration-dependent information using the Benchmark Concentration (BMC) modeling approach. The generated results support the potential applicability of CPP for both primary screening (i.e., generating

and clustering of compound activity profiles based on a diverse set of features) and secondary screening (i.e., dissecting the compound's potency and potential MoA in more detail). We therefore would not limit the application of CPP to either primary or secondary screenings, but instead leave this decision to the researchers looking for the most convenient assay to fulfill their research goals. We addressed this aspect in the revised manuscript as follows:

- Discussion: *“In addition, the approach taken in this study to evaluate the CPP method by using a set of defined reference compounds at multiple, partially cytotoxic, concentrations aimed at comparing CPP against published CP data. Although this approach can enable the evaluation of transferability CPP within and between labs, it was not intended to be transferred to large-scale HTPP of hundreds of thousands of compounds. In fact, depending on the site- and screen-specific requirements, unnecessary costs can be avoided by considering already available compound- and cell type-specific cytotoxicity information or conducting cytotoxicity pre-screens to identify effective but not overly cytotoxic concentration ranges to be used for subsequent phenotypic profiling using CPP.”*

Specific points:

1.) QUOTE: *“116 All CPP dyes were well detectable over the course of 4 weeks after staining, with some fluctuations in the staining intensities observed for the Lyso and ER dyes from day 2 after staining (Supplementary Fig. 1B).”*

A fluctuation is a change between alternating states. As such, the temporal change observed for the ER stain cannot be handwaved as a “fluctuation”. Rather, it appears to be a sustained and stable increase. If this is a real observation, it needs to be better explained/discussed.

Answer:

We thank the reviewer for this comment. To our knowledge, differences in the brightness of dyes over time have also been observed by other experienced CP users in the community [personal communication] and the pH dependency of the ConA ER dye in particular has already been discussed in the CP wiki (<https://github.com/carpenter-singh->

[lab/2023_Cimini_NatureProtocols/wiki](https://doi.org/10.1038/nprot.2023.10)). In our revised manuscript we elaborate on that aspect as follows:

- **Results:** *“All CPP dyes were well detectable over the course of 4 weeks after staining, but the staining intensities of all dyes remained sufficiently stable only until day 1 (deviation of less than $\pm 10\%$ compared to day 0) (Supplementary Fig. 1A). Prominent differences in the relative signal intensities over time were observed for the Lyso dye (decreasing) and the ER dye (increasing) already from day 2 after staining. This may be related to pH-dependent changes in the fluorescence properties of the dye itself, or due to changes in the binding of the organelle-targeting moiety. The LysoTracker™ dye accumulates in lysosomes due to their acidic pH (~4.5-5.0) and is therefore being applied to live cells. In turn, concanavalin A is used in fixed cells and binds to specific carbohydrate structures, such as mannose and glucose residues, that are enriched at the ER. After cell fixation by cross-linking proteins and other cellular components with paraformaldehyde (PFA), the cellular morphology is preserved, and the ability of fluorescent dyes to diffuse within the cell should be significantly limited, but is not fully inhibited. Under the staining conditions used in this study, the Lyso and ER dyes may take longer (up to day 2) to fully reach equilibrium. Whereas the signal intensity of the ER dye is maintained at this plateau, the intensity of Lyso may have quickly dropped again to lower levels after day 2. This emphasizes that thoroughly characterizing the fluorescence properties of the dyes used in the specific experimental setting is crucial for interpreting the results, especially when using dyes in different cellular compartments with distinct pH levels, such as in the highly acidic environment of lysosomes. For that reason, imaging was conducted in the CPP assay within 24 hours after staining to ensure robustness of phenotypic profiling data.”*

2.) QUOTE: *“186 The image analysis procedure included the registration and segmentation of images (Supplementary Fig. 2A), followed by the feature extraction (Supplementary 188 table 3, 5) using either the commercial Harmony software (Revvity Inc.) (894 features) or the open-source Cell Profiler software (3,044features).”*

The fact that the choice of feature extraction software is not made explicit throughout the manuscript makes it ambiguous at times. Also, since the authors went through the trouble of

using both feature extraction methods, it would be good to compare the performance of the two. As someone who has access to both, this reviewer would for one appreciate an analysis demonstrating whether one would be superior to the other. If they lend themselves equally well to the methodology, that would be good to know as well.

Answer:

We agree with the reviewer. Throughout this study, we generally used Harmony by default for all analyses unless otherwise specified. In the revised manuscript and figures, we more clearly indicate, when we used the Harmony software or Cell Profiler. In particular, we clarified in the manuscript:

- **Results: *“Using this reference compound plate, we conducted small-scale CPP screens with 48 hours exposure time in MCF-7, HepG2, U2OS, and RPTEC cells. For direct comparison of the CPP profiling data with CP, we also conducted the reference compound screen using the CP method in MCF 7 cells. The data was analyzed using customized image and data analysis workflows as outlined for CPP in Fig. 2B. The CPP and CP image analysis procedures included the registration and segmentation of images (Supplementary Fig. 2A), followed by the extraction of cell features using the commercial Harmony software (Revvity Inc.) (CPP/Harmony: 894 features, CP/Harmony: 558 features), representing the primary image analysis software that was applied to all cell lines in this study. For comparison, we additionally analyzed CPP in MCF-7 cells in an analogous way using the open-source Cell Profiler software (CPP/Cell Profiler: 3,648 features) as an alternative image analysis method.”***

We have now further included a more detailed comparison of CPP results obtained when using Harmony vs Cell Profiler based on Spearman correlation and hierarchical clustering (see our response to the specific comment 13 from reviewer 3), referred to as profile similarity plots (see Fig. 5C). Based on this comparison, we added to the conclusion:

- **Discussion: *“Furthermore, the direct comparison of the profile similarity plots obtained when using the Harmony or Cell Profiler software for feature extraction showed that often higher feature correlation values were obtained among compounds with the same annotated MoA when using Harmony. Although both methods led to the clustering of the same compounds into the “Mito cluster”,***

Harmony provided a better separation of the other compounds, e.g., into a more distinct “Lyso cluster” comprising all three compounds with annotated Lyso activities. Moreover, from the large number of features being extracted when using Cell Profiler, ~600 features had to be filtered out because of a high number of missing values. This made the use of Cell Profiler less convenient, as it required more extensive initial filtering of features and may consequently produce different feature lists for individual experiments. Previous reports from several laboratories [Chandrasekaran, 2024; Rohban, 2017; Way, 2022] have also noted the need to filter and reduce features when using Cell Profiler and with our analysis we concur with these findings. In turn, the selected features from Harmony proved reliable for downstream analysis without the need for feature filtering and reduction, facilitating more consistent comparisons across different screens and laboratories.”

3.) QUOTE: “209 profiles for the DNA and Mito channels (Supplementary Fig. 2C). However, the activity profiles of fluphenazine and tetrandrine indicated clear Mito activities in CP but not 211 CPP (Fig. 2C-D, Supplementary Fig. 2C). Although those Mito activities were consistent with published CP profiles of MCF-7 cells, they may not represent the most relevant activity when considering the reported, primarily Lyso-related MoA for the two compounds (Fig. 2A).”

This claim seems unsubstantiated. Just because the compounds have lyso- activities reported doesn’t necessarily mean that they do not have mito- activity. Rather than attempt to explain why CP mito captures an event and CPP mito doesn’t, the authors have opted to discount the possibility that these compounds have mito- effects. This warrants further investigation to understand what the ground truth here is (is there a mito- effect or not?), and then decide based on that which method is capturing the ground truth better.

Answer:

We thank the reviewer for this very valuable feedback. We indeed cannot be sure whether the “true” impact of these two compounds on mitochondria is more relevant than on lysosomes. Importantly, in the initial figures, the seemingly missing response in the Mito channel when using CPP was a consequence of a not ideal selection of the scaling/color-coding of the compound activity profile plots. In the initial heatmaps, the number of bins

used was just too small to adequately represent the data, which made the existing Mito effect from CPP data visually unclear. In the revised figures, we have now updated all compound activity heatmaps using two more bins, making the effect clearer. Consequently, we have also revised the statement the reviewer refers to as follows:

- Results: *“Overall, CPP and CP showed largely consistent compound activity profiles for the DNA and Mito channels common to both methods (Supplementary Fig. 2C-C’), with the Mito activity profiles of fluphenazine and tetrandrine showing higher robust z-score magnitudes in CP than in CPP (Fig. 2C-D, Supplementary Fig. 2C).”*

4.) QUOTE: *“173 Barcoding compound effects using CPP improves the specificity of phenotypic profiles”*

This claim appears to be unsubstantiated by the data and mostly bolstered by the use of compounds with lyso-related MoA. It is not clear from the current data that CPP would be better at large-scale comparisons with compounds of all MoAs.

Answer:

We agree with the reviewer that at this point in the manuscript, additional evidence would be needed to support this broad statement. We have therefore changed the section title to:

- Results/Subheading 3: *“Barcoding compound effects using CPP expands the diversity of phenotypic profiles”*

Importantly, as described in the revised manuscript, we have now performed the suggested large-scale comparisons with compounds of all MoAs using spearman correlation and hierarchical clustering of the feature profiles of CP and CPP (see Fig. 5C, Fig. S6C, Fig. S7C) (see our response to the specific comment 13 from reviewer 3). To us, this additional analysis shows that CPP provides better cluster resolution and provides support that, e.g., separating Actin and Golgi can be beneficial to enhance the organelle-specificity of phenotypic profiles for comparative MoA analyses. In addition, in the subsequent responses we describe that separating RNA and ER provides qualitatively different information in the cytoplasmic region (see our response to the specific comment 7 from reviewer 3) and that a channel that combines stains for two components does not necessarily reflect the combined

effects on both components at the level of BMC modeling (see our response to the specific comment 9 from reviewer 3).

5.) QUOTE: “248 increased data variability to some extent. To facilitate analysis of the BMC modeling data, we further assigned each of the 894 (CPP) or 558 (CP) features to 62 (CPP) or (CP) biologically meaningful feature categories representing specific combinations of channels, cell regions, and analysis modules (Supplementary table 5; Supplementary table 6) in a similar way as previously described for CP. “

The 894 and 558 are extracted how? Harmony? Or Cell Profiler?

Answer:

We used the Harmony software to extract the 894 (CPP) and 558 (CP) features. We clarified that in the revised the manuscript (see our response to the specific comment 2 of reviewer 3) as follows:

- **Results: “The CPP and CP image analysis procedures included the registration and segmentation of images (Supplementary Fig. 2A), followed by the extraction of cell features using the commercial Harmony software (Revvity Inc.) (CPP/Harmony: 894 features, CP/Harmony: 558 features), representing the primary image analysis software that was applied to all cell lines in this study. For comparison, we additionally analyzed CPP in MCF-7 cells in an analogous way using the open-source Cell Profiler software (CPP/Cell Profiler: 3,648 features) as an alternative image analysis method.”**

6.) QUOTE: “281 To further distinguish between early (most sensitive, potentially primary) and later (less sensitive, potentially secondary) compound effects, we further investigated the specific sequence of feature category responses as well as the maximum sizes of those effects for CPP (Fig. 3D; Supplementary Fig. 4A-O) and CP (Supplementary Fig. 5A-O) using accumulation and magnitude plots as previously described for CP.”

This is confusing because it refers to temporal effects (early vs late) but instead, the actual analysis is concentration, not time-based. Perhaps the authors are referring to “specific/low

concentration” vs “non-specific/high concentration” effects here? Either way, calling it early/late might be confusing to readers.

Answer:

We agree with the reviewer. Now we have changed all occurrences of using “early” or “late” in the revised manuscript to “low-concentration” or “high-concentration”.

- **Results:** *“To further distinguish between low-concentration (most sensitive, potentially primary) and high-concentration (less sensitive, potentially secondary) compound effects, we further investigated the specific sequence of feature category responses as well as the maximum sizes of those effects for CPP (Fig. 3D; Supplementary Fig. 4A-O) and CP (Supplementary Fig. 5A-O) using accumulation and magnitude plots as previously described for CP.*

7.) QUOTE: *“294 with its broad activity shown in the Proportion BMC profile (Fig. 3B). In particular, CPP was able to distinguish between RNA and ER features showing maximum effect sizes, highlighting the advantage of separating RNA and ER channels to more precisely determine different MoA.”*

When RNA and ER are separated by cell compartment (not just by channel), CP appears to capture distinct changes in RNA and ER as well (Fig 3B “Channels + Regions”).

Answer:

We thank the reviewer for the important comment. We agree that in Fig. 3B we can observe that the channel RNA/ER has different profiles depending of the compartment (cytoplasm vs nucleus) in CP. This observation is also reflected in the following BMC modeling plots (Fig. R1 and Fig. R2) for two exemplary features of MCF-7 cells treated with fulvestrant (see Fig. 3A’ of the manuscript).

Figure R1 shows an example of a feature that had different profiles in the cytoplasm compared with the nucleus, which was consistent for both CPP and CP. In the cytoplasm of the separated RNA channel (CPP) the calculated BMC was 0.0011 μM and in the merged RNA/ER channel (CP) it was 0.00049 μM . However, in both CPP and CP no BMC (BMC^{NA}) could be calculated for the nucleus region indicating no activity, which was also the case for

the separated ER channel (CPP) in the cytoplasmic region. This means that, for this particular feature, fulvestrant seems to affect only RNA in the cytoplasm but neither RNA in the nucleus nor ER in the cytoplasm. The different RNA profiles in these compartments could be due to different levels of RNA in the nucleus and the cytoplasm since RNA accumulation/degradation may be different in these two compartments.

Fig. R1

Figure R1. Plot showing BMC fittings for the feature “Intensity_[...] Mean” in the cytoplasm (left column) and nucleus (middle column) of the separated CPP RNA (Alexa 488, upper row) and the merged CP RNA/ER (Alexa 488, lower row) channels of cells treated with fulvestrant. The upper right plot shows the BMC fitting for the same feature in the cytoplasm in the separated CPP ER (Alexa 647_2) channel. Features for which no BMC could be determined (BMC^{NA}) or with estimated BMCs greater than the highest tested non-cytotoxic concentration (BMC^{HIGH}) were considered inactive. Further details about the BMC modeling are described in the legend to Fig. 3A’ in the manuscript or in the SI.

However, we are not of the opinion that this means that CP can generally detect RNA-related responses in the cytoplasm independently of ER-related responses, because such effects detected in the merged RNA/ER channel could also be due to ER-related effects in

the cytoplasm. This scenario is illustrated by the following BMC modeling plots (Fig. R2) for another exemplary feature of MCF-7 cells treated with fulvestrant (see Fig. 3A' of the manuscript).

Fig. R2

Figure R2. Plot showing BMC fittings for the feature “Morphology_STAR_ [...] Symmetry 04” in the cytoplasm (left column) and nucleus (middle column) of the separated CPP RNA (Alexa 488, upper row) and the merged CP RNA/ER (Alexa 488, lower row) channels of cells treated with fulvestrant. The upper right plot shows the BMC fitting for the same feature in the cytoplasm in the separated CPP ER (Alexa 647_2) channel. Features for which no BMC could be determined (BMC^{NA}) or with estimated BMCs greater than the highest tested non-cytotoxic concentration (BMC^{HIGH}) were considered inactive. Further details about the BMC modeling are described in the legend to Fig. 3A' in the manuscript or in the SI.

Opposite to the feature shown in Fig. R1, this feature showed concordant responses in the nucleus of both the separated RNA channel (CPP) with a calculated BMC of 0.00062 μ M and in the merged RNA/ER channel (CP) with a calculated BMC of 0.0014 μ M. However, the responses of CPP and CP were not concordant for the same channels in the cytoplasmic region. For the separated RNA channel (CPP), the calculated BMC was greater than the

highest tested non-cytotoxic concentration (BMC^{HIGH}) in the cytoplasm indicating no activity of that particular feature. In contrast, for the merged RNA/ER channel (CP), a BMC of 0.0008 μM was calculated indicating activity. This seemingly contradictory results for the RNA channel in the cytoplasmic region could only be explained when specifically considering responses in the cytoplasm of the separated ER channel (CPP), in which this feature also showed clear activities at a very similar concentration ($BMC = 0.00095 \mu M$). This example shows that, when using CP, this more ER-specific response would have been misinterpreted. Whereas, CP cannot resolve these differences, CPP can detect changes in profiles independently for RNA and ER. This illustrates the power of CPP to separate the RNA and ER channels.

To address the question in the revised manuscript, if RNA and ER provide qualitatively similar or different information in the cytoplasmic region, we are now directly comparing in the new Fig. S2C' (right plot) all the features extracted only from the cytoplasmic region of the RNA or ER channels from CPP with the merged RNA/ER channel from CP in a scatter plot. The x-axis corresponds to the robust z-scores for CP and the y-axis the corresponding values for CPP.

Fig. S2C'

Figure S2C'. Scatter plots comparing robust z-scores of all features from the Generic, DNA, and Mito channels that are common in CPP and CP (left plot), or only the cytoplasm features from the separated RNA and ER channels in CPP with the merged RNA/ER channel in CP (right plot). The x-axis corresponds to the robust z-scores for CP and the y-axis the corresponding values for CPP. Pearson correlation is performed using the robust z-score of each feature (feature-level). Further details about the plot are given in the legend to Fig. S2C' in the SI.

The correlation plot shown in Fig. S2C' (right) supports our statement that separating these channels provides qualitatively different information. In the revised manuscript, we refer to this as follows:

- Results: *“This observation is one example of the usefulness of separating the ER and RNA channels, which provides qualitatively different information. This is further supported by the direct comparison of the features extracted for all compounds from only the cytoplasmic region of the RNA and ER channels (CPP) with the merged RNA/ER channel (CP) (Supplementary Fig. 2C’).”*

8.) QUOTE: *“Rapamycin, a highly specific mTOR inhibitor, caused a strong effect on the number of nucleoli per 317 cell (Fig. 4A; Supplementary Fig. 4I), consistent with the role of the mTOR pathway in controlling cell cycle progression through regulating ribosome biogenesis and nucleoli numbers.”*

The cellular IC50 for mTOR is in the low nanomolar range. The tested concentration of 10 μ M is too high, and more so for a 48 hours assay. It is known that prolonged treatment with Rapamycin (after 24 hours) has inhibitory effects on mTORC2 activity.

Answer:

For rapamycin, we used a concentration from 10 μ M to 0.00318 μ M as described in Supplementary table 2. With regard to the observed effects of rapamycin on nucleoli, we would like to refer the reviewer to the legend of Fig. 4A, where we stated that *“Representative images for highest non-cytotoxic concentrations are shown (asterisk) and compared to the DMSO solvent control.”* We intended to show in Fig. 4A compound effects on cells consistently at maximum concentrations/effect levels as indicated by the asterisk below the compound activity profiles (which are close-ups from the compound activity profiles described in Fig. 2C and Fig. S2B). We hope that this becomes clearer now in the revised Fig. 4A and the corresponding updated figure legend.

- Figure legend/Fig. 4: *“Representative images for highest non-cytotoxic concentrations (as indicated by an asterisk below the corresponding concentration under each heatmap profile) are shown and compared to the DMSO solvent control.”*

Fig. 4A

Figure 4A. Representative images (RNA channel) and corresponding compound activity profiles for nucleoli-related features (part of profiles described in Fig. 2C and Fig. S2B) showing activities of four exemplary reference compounds on the two feature categories *RNA Nucleoli Number* (comprising three features) and *RNA Nucleoli Morphology* (comprising five features) at eight tested concentrations (increasing from left to right, dashed outlines indicating cytotoxic ranges) across four different cell lines. Further details about the plot are given in the legend to Fig. 4A in the manuscript.

The heatmap indicates that the phenotypic response of nucleoli features to rapamycin is consistently pronounced across the entire tested concentration range. Likewise, the BMCs for all rapamycin features are calculated using all the concentration range and not only 10 μM . We would like to emphasize that we see an effect of rapamycin on nucleoli numbers already at the lowest tested concentration of 3 nM (see Fig. 4A, Fig. 4C), which is in line with the published IC_{50} range. As the lowest tested concentration was still too high for calculating reliable BMCs for some feature categories, we expect that some cellular responses will be observed even in the sub-nanomolar range. We made that point more explicit in the manuscript text as follows:

- Results: “*The compound activity plots (Fig. 4A) and BMC accumulation plots (Fig. 4C) revealed that rapamycin showed activities already at lowest concentrations (3 nM) among the tested reference compounds in MCF-7, HepG2 and U2OS cells, but interestingly no relevant responses in RPTEC cells. As the lowest tested concentration of rapamycin was still too high for calculating a reliable BMC for the nucleoli number feature category, it was set to the next lower concentration (as described in the*

Supplementary information). In fact, this suggests that nucleoli responses to rapamycin will be observed even in the sub-nanomolar range.”.

9.) QUOTE: “412 However, when using the CP method, these compounds exhibited pronounced responses in the combined RNA/ER channel (Supplementary Fig. 5E, L, O), indicating that the morphological changes in the RNA/ER channel in the CP method mainly reflect those in the ER channel in the CPP assay, highlighting the advantage of separating RNA and ER channels to more precisely determine different MoA”

This is stating the obvious – a channel that combines stains for two components will show the combined effects on both components. The question should be whether adding compartment analysis to this channel (nucleus vs cytoplasm) in CP would distinguish between effects on RNA vs effects on ER. In this case, this advantage is not entirely lacking from CP.

Answer:

As responded to specific comment 7 of reviewer 3, there are scenarios when CP will not be able to separate RNA vs ER using different compartments. In CP, when we observed changes in the RNA/ER channel specifically in the cytoplasm, we cannot determine whether this is due to an effect on the ER located in the cytoplasm or a specific change in RNA levels in the cytoplasm compared to the nucleus.

In addition, the reviewer is assuming that “a channel that combines stains for two components will show the combined effects on both components”, which we think is not necessarily true at the level of BMC estimation as illustrated by the following two BMC modeling plots for exemplary features of MCF-7 cells treated with fulvestrant (Fig. R3) or cytochalasin D (Fig. R4). Those plots show two example features, for which BMCs were calculated only in one CPP channel, whereas CP was not able to detect this BMC in the combined channel. In the first example shown in Fig. R3, the feature showed a response (BMC = 0.0012 μ M) in the cytoplasm of the separated ER channel (CPP) but no response in the separated RNA channel (CPP) or in the merged RNA/ER channel (CP). Figure R4 shows a similar example but in the ring region of the separated Golgi and Actin channel (CPP) and the merged AGP channel (CP). This illustrates the power of separating the channels in CPP when calculating BMCs. This further shows that a channel that combines stains for two

components does not necessarily reflect the combined effects on both components at the level of BMC modeling. We largely re-wrote the section that the reviewer mentions in this comment and would like to refer the reviewer to the revised manuscript and our responses to the specific comment 13 of reviewer 3 for further details.

Fig. R3

Figure R3. Plot showing BMC fittings for the feature “Texture_SER_2_ [...] SER Bright 2 px” in the cytoplasm of the separated CPP RNA (Alexa 488, left) and ER (Alexa 647_2, middle), and the merged CP RNA/ER (Alexa 488, right) channels of cells treated with fulvestrant. Features for which no BMC could be determined (BMC^{NA}) or with estimated BMCs greater than the highest tested non-cytotoxic concentration (BMC^{HIGH}) were considered inactive. Further details about the BMC modeling are described in the legend to Fig. 3A’ in the manuscript or in the SI.

Fig. R4

Figure R4. Plot showing BMC fittings for the feature “Texture_Haralick_1_ [...] Haralick Correlation 1 px” in the ring region of the separated CPP Golgi (Alexa 488_2, left) and Actin (HOECHST 33342, middle), and the merged CP AGP (Alexa 568, right) channels of cells treated with cytochalasin D. Features for which no BMC could be determined (BMC^{NA}) or with estimated BMCs greater than the highest tested non-cytotoxic concentration

(BMC^{HIGH}) were considered inactive. Further details about the BMC modeling are described in the legend to Fig. 3A' in the manuscript or in the SI.

10.) QUOTE: “423 In this proof-of-concept study, we established an efficient, robust, and broadly applicable HTPP assay for multiplexed, iterative staining of single cells with at least seven fluorescent dyes labeling at least nine different subcellular compartments and organelles.”

Per previous comments above, this review does not agree with this. If anything, the method was de-optimized for HTPP. Rather, it was optimized for having more explanatory and mechanistic power than CP (maybe), but the trade-off against HTPP is quite large.

Answer:

As responded in greater detail in our response to the general comments of reviewer 3, we agree that CPP is optimized for having more explanatory power, with the trade-off of having a second round of staining. In the revised manuscript, we updated this part as follows:

- Discussion: *“In this proof-of-concept study, we established a customizable, robust, and broadly applicable phenotypic screening assay for multiplexed, iterative staining of single cells with at least seven fluorescent dyes labelling at least nine different subcellular compartments and organelles. The CPP method thus adds a new approach to the versatile HTPP methods and offers additional options for addressing MoA-specific research questions.”*

In a following part of the discussion, we further address the comment that “the method was de-optimized for HTPP” as follows:

- Discussion: *“With developing the CPP method, we aimed to provide the growing Cell Painting community with a more customizable version of the very successful CP approach with increased organelle-specificity of the phenotypic profiles, accepting the trade-off that this optimization reduces the high-throughput capacity of CPP compared to CP. Since not every research question may require the separation of the AGP and RNA/ER channel or addition of the Lyso channel, it will be up to the individual scientist to decide, if the advantages of CPP serve their specific research*

needs and outweigh a higher investment in cost, time, and resources compared to CP.” (this part is also covering the aspect on cost/labor discussed next)

11.) QUOTE: “489 It is further important to consider that, in comparison to CP, the CPP assay necessitates (at least) one additional elution, staining, and imaging cycle, thereby demanding larger data storage and high-performance computing capacity. However, working with cells at higher confluency would reduce the number of imaging fields and thus also the required amount of imaging data.”

Imaging fewer fields does not counteract the added burden of liquid handling and reagent costs. It might tradeoff statistical robustness.

Answer:

We agree with the first part of this specific comment that “Imaging fewer fields does not counteract the added burden of liquid handling and reagent costs.”. As answered in greater detail in our response to the general comments of reviewer 3, we address this aspect in another paragraph that was added to the revised manuscript as follows:

- Discussion: ***“With developing the CPP method, we aimed to provide the growing Cell Painting community with a more customizable version of the very successful CP approach with increased organelle-specificity of the phenotypic profiles, accepting the trade-off that this optimization reduces the high-throughput capacity of CPP compared to CP. Since not every research question may require the separation of the AGP and RNA/ER channel or addition of the Lyso channel, it will be up to the individual scientist to decide, if the advantages of CPP serve their specific research needs and outweigh a higher investment in cost, time, and resources compared to CP.”***

To address the second part of this specific comment that “It might tradeoff statistical robustness.”, we now checked how the statistical robustness is affected by different field of view vs different number of cells. Since we have performed the CP/ CPP experiments using 2 fields of view, we used that data and conducted a resampling method, where n cells were randomly selected per each compoundXconcentration from a single or both fields of view (see Fig. S1F).

Fig. S1F

F Influence of the number of cells and imaging fields at 20x magnification on the variance of feature data in CP and CPP

Figure S1F. Evaluation of the influence of the number of cells and imaging fields at 20x magnification on the variance of feature data in CP and CPP. Plots show the median coefficient of variation (CV) across wells and plates of unprocessed, Harmony-extracted CP or CPP features depending on the number of fields and number of randomly chosen cells.

Our results show that there is no tradeoff in statistical robustness related with the number of fields being imaged but that data quality is mainly proportional to cell number, which is concordant with the conclusions by Cimini et al., 2023. We address this aspect in the revised manuscript as follows:

- Results: *“Our CP and CPP analyses further show that the coefficient of variation of feature data between cells depends mainly on the number of cells being imaged but not on the number of imaging fields per se, indicating that capturing ~ 2,500 cells is a sufficient number to ensure statistical robustness of the data (Supplementary Fig. 1F).”*
- Discussion: *“However, working with cells at higher confluency would reduce the number of imaging fields, without obvious confounding effects on the statistical robustness of the feature data (Supplementary Fig. 1F), and thus also the required amount of imaging data. “*

12.) *The creation of CPP from CP is enabled by the elution buffer, which the authors claim is superior to other elution buffers for this application. Thus, the development of this buffer is a central technological component of this method. However, details about this buffer, its composition, and the process by which it was selected and optimized are completely lacking in the manuscript.*

Answer:

We agree that the elution buffer is a central part of the CPP assay. In the initially submitted manuscript, we already summarized the development of the elution buffer (and alternate dye-specific elution buffer variants) as follows: “*The development and optimization of suitable elution buffers for each dye involved extensive testing of various buffer components and parameters in combination including different pH, reducing agents, chaotropic agents (ionic strength), temperatures, and elution times. The optimal elution buffer compositions for each dye are summarized in Supplementary table 1 to guide other laboratories in implementing and customizing their own CPP assay depending on their specific needs.*”. We understand that the full composition of the elution buffer used in this study should not only be described in the Table S1 and the SI. In the revised manuscript, we therefore added a statement on its composition to the section “CPP enables iterative staining-elution cycles using an efficient dye elution buffer” as follows:

- **Results: “*Notably, the CPP elution buffer (0.5 M L-Glycine, 1% SDS, pH 2.5), which was used in this study for all phenotypic profiling screens, was designed to efficiently remove the signals of all dyes except for the Mito dye (Fig. 1C).*”.**

13.) *The authors attempt to give the manuscript a mechanistic/discovery angle, but the observations are often shallow and unsubstantiated by the data presented. In-depth mechanistic investigations are lacking for observations to be meaningful.*

Answer:

Thank you very much for raising this important aspect. As already mentioned in responses to other comments above, we further improved the respective sections by adding more in-depth analyses of our data to strengthen our mechanistic conclusions.

We understand that this specific comment of reviewer 3 mainly relates to the capability of CPP to identify distinct alterations in morphological Actin and Golgi vs AGP as well as RNA and ER vs RNA/ER features, which cannot be distinguished in the same cell regions by CP.

These new analyses include the generation of profile similarity plots (robust z-score, feature-level, maximum non-cytotoxic compound concentration) for each cell line using Spearman correlation and hierarchical clustering, which enabled

- a direct comparison of the overall compound activity profiles (shown in Fig. 2, Fig. S2) of all reference compounds with each other,
- the investigation of different responses between the four cell lines,
- the investigation of differences between CPP vs CP in MCF-7 cells, and
- the investigation of differences between Harmony vs Cell Profiler in MCF-7 cells.

These new analyses were included in our case studies on cytochalasin D/latrunculin B (Use case 2; shown in Fig. 5, Fig. S6) and sunitinib malate/siramesine/tetrandrine/fluphenazine (Use case 3; shown in Fig. 6, Fig. S7-10). In those profile similarity plots, we correctly identified the clustering of reference compounds with annotated Mito- or Lyso-related MoAs, respectively, in all cell lines except RPTEC (see Fig. S7C).

Fig. S7C

Profile similarity plots (robust z-score, feature-level, including Lyso features) of all reference compounds at highest non-cytotoxic concentrations in different cell lines

Figure S7C. Profile similarity plots of all reference compounds at highest non-cytotoxic concentrations generated by Spearman correlation (CPP/Harmony, robust z-scores, feature level) of all features (including Lyso features), and hierarchical clustering across four different cell lines. Negative control compounds (i.e.,

saccharine, sorbitol) are distinguished by slight transparency. Hierarchical clusters are visualized in the grey bar below the matrices by different shades of grey. Compounds annotated to either Actin- (cyan), Mito- (orange), or Lyso- related (red) MoAs (see Fig. 2A) are labelled by color-coded names or dashed outlines in the matrix. More details are described in the legend to Fig. 5C in the manuscript or in the SI.

The profile similarity plots further revealed a high correlation and clustering of sunitinib malate - a multi-targeted tyrosine kinase inhibitor - with the three “Lyso cluster” compounds tetrandrine, fluphenazine, and siramesine, which was particularly evident in MCF-7 and U2OS cells. This indicated a mechanistic similarity to the three compounds with annotated Lyso-related MoA, with Lyso responses occurring rather as secondary effects as suggested by the corresponding BMC plots (see Fig. 6C).

Interestingly, despite their common Actin-related MoA, cytochalasin D and latrunculin B did not show a high phenotypic profile similarity and, thus, did not form a discrete cluster in most cell lines (see Fig. S7C). The corresponding BMC plots further showed that although both compounds induce similar Actin responses in terms of the total number of affected Actin features (see Supplemental material 18, left plot), the total number of affected Golgi features (see Supplemental material 18, right plot) clearly differed. In fact, the correlation of the overall compound activity profiles of latrunculin B was higher to the microtubule inhibitor nocodazole than to cytochalasin D in MCF-7, HepG2 and U2OS cells (see Fig. S7C), which was also supported by the corresponding BMC plots (see Fig. R5 and Supplemental material 18, left and right plot). This example demonstrates the utility of the profile similarity plots for identifying compounds with similar phenotypic profiles, and of the BMC method for more detailed analyses of concentration-dependent phenotypic responses and MoA dissection.

Fig. R5

Figure R5. BMC bar plots of cytochalasin D (top), latrunculin B (middle), and nocodazole (bottom) showing AGP (CP) vs Actin and Golgi (CPP) responses at the BMC level. The BMC bar plots show the total number of in-/active BMC features from the cytoplasm, ring region, and membrane region that are related to AGP (CP), Actin (CPP), or Golgi (CPP) channels in MCF-7 cells. The right plots show the same data but grouped for different concentration ranges. The corresponding BMC bar plots of all reference compounds are available as Supplementary material 18 in the SI.

We further found that the clustering of cytochalasin D clearly differed between CPP and CP in MCF-7 cells (see Fig. 5C). To enable direct comparison of the compound activity profiles of CPP and CP in MCF-7 cells, the Lyso features were excluded from this analysis.

Fig. 5C

Profile similarity plots (robust z-score, feature-level, **excluding Lyso** features) of all reference compounds at highest non-cytotoxic concentrations in MCF-7 cells

Figure 5C. Profile similarity plots of all reference compounds at highest non-cytotoxic concentrations in MCF7 cells generated by Spearman correlation (robust z-scores, feature level) of all except Lyso features, and hierarchical clustering. Analyzed feature data are either generated in CPP (CPP/Harmony, CPP/CellProfiler) or CP (CP/Harmony), or extracted from different image analysis software (CPP/Harmony, CPP/Cell Profiler). Negative control compounds (i.e., saccharine, sorbitol) are distinguished by slight transparency. Hierarchical clusters are visualized in the grey bar below the matrices by different shades of grey. Compounds annotated to either Actin- (cyan), Mito- (orange), or Lyso- related (red) MoAs (see Fig. 2A) are labelled by color-coded names or dashed outlines in the matrix. More details are described in the legend to Fig. 5C in the manuscript or in the SI.

In CP, cytochalasin D was part of the “Lyso cluster” in the profile similarity plots, with a high correlation to fluphenazine and tetrandrine that both exerted AGP and prominent RNA/ER responses (see Fig. R6, Fig. R7, and Supplemental material 18 and 19). Similarly, cytochalasin D also showed considerable responses in those two CP channels, which contributed to the observed clustering. When using CPP, those responses of fluphenazine and tetrandrine could be more clearly separated into stronger Golgi than Actin and stronger ER than RNA responses (see Fig. R6, Fig. R7, and Supplemental material 18 and 19). Importantly, in CPP, cytochalasin D showed much stronger Actin than Golgi responses, which contributed to its clear separation from the other compounds in the profile similarity plots (see Fig. 5C). Together, this example demonstrates that the separation of RNA/ER and AGP responses that is achieved by CPP can better resolve differences in compound responses.

Fig. R6

Figure R6. BMC bar plots of cytochalasin D (top), fluphenazine (middle), and tetradrine (bottom) showing AGP (CP) vs Actin and Golgi (CPP) responses at the BMC level. The BMC bar plots show the total number of in-/active BMC features from the cytoplasm, ring region, and membrane region that are related to AGP (CP), Actin (CPP), or Golgi (CPP) channels in MCF-7 cells. The right plots show the same data but grouped for different concentration ranges. The corresponding BMC bar plots of all reference compounds are available as Supplementary material 18 in the SI.

Fig. R7

Figure R7. BMC bar plots of cytochalasin D (top), fluphenazine (middle), and tetrandrine (bottom) showing RNA/ER (CP) vs RNA and ER (CPP) responses at the BMC level. The BMC bar plots show the total number of in-/active BMC features from the cytoplasm that are related to RNA/ER (CP), RNA (CPP), or ER (CPP) channels in MCF-7 cells. The right plots show the same data but grouped for different concentration ranges. The corresponding BMC bar plots of all reference compounds are available as Supplementary material 19 in the SI.

To include these new more detailed MoA analyses, we largely re-wrote the sections about case studies 2 and 3 in the revised manuscript to further substantiate the MoA aspect. In that regard, we added the following concluding statement to the revised manuscript:

- Discussion: *“The case studies indeed suggested that separating Actin and Golgi as well as ER and RNA channels and, in particular, addition of the Lyso channels may be beneficial to enhance the organelle-specificity and diversity of phenotypic profiles for comparative MoA analyses. However, the number of compounds for different MoA provided in this study was yet limited and the first indications regarding a potential higher phenotypic profile-specificity of CPP over CP will need to be strengthened by comparing the reference compound profiles against a larger set of compounds with known MoA.”*

As already mentioned in our response to the specific comment 2 from reviewer 3, we have also used the profile similarity plots for a more detailed comparison of CPP results obtained when using Harmony vs Cell Profiler (see Fig. 5C). Based on the results of this comparison, we added the following concluding statement to the revised manuscript:

- Discussion: *“Furthermore, the direct comparison of the profile similarity plots obtained when using the Harmony or Cell Profiler software for feature extraction showed that often higher feature correlation values were obtained among compounds with the same annotated MoA when using Harmony. Although both methods led to the clustering of the same compounds into the “Mito cluster”, Harmony provided a better separation of the other compounds, e.g., into a more distinct “Lyso cluster” comprising all three compounds with annotated Lyso activities.”*

Response to reviewer comments on von Coburg et al.

“Cell Painting PLUS: Expanding the multiplexing capacity of Cell Painting-based phenotypic profiling using iterative staining-elution cycles”

Comments received by authors: 13 Dec 2024

Response submitted by authors: 24 JAN 2025

Thank you again for the very constructive feedback from the reviewers. We have revised the manuscript according to their remaining suggestions.

Reviewer #1:

Specific points:

We thank the author for their response to our initial criticisms. However, their responses about software are somewhat troubling - not performing a feature reduction step on CellProfiler features is not the accepted way to perform the assay (Bray 2016, Caicedo 2017, Cimini 2023, in addition to the authors own citations of Chandrasekaran, Rohban, and Way), so any comparisons between CellProfiler and Harmony become immediately suspect when the CellProfiler feature set is not used per long-established best practices. Feature reduction is a very simple procedure which can, if maximum comparability across two data sets taken in two locations is desired, be performed in a single line of code by simply importing the column list from one data set to the other. Such a simple transition is notably NOT possible if one location has Harmony and one does not because it uses another brand of microscope, since most if not all high-content imagers can successfully perform Cell Painting (Bray 2016, Cimini 2023, Tromans-Coia 2023).

Answer:

We understand the concerns of the reviewer and agree with the suggestion that the comparison between Cell Profiler and Harmony would be more robust when the Cell Profiler feature set is analyzed including a feature selection/reduction step in addition to the basic data cleaning process.

Therefore, we performed feature reduction for the Cell Profiler data by using the Pycytominer software (Serrano et al., 2024), which has already been used to process the JUMP and Cell Painting Gallery datasets. We updated the corresponding profile similarity plot (Fig. 5C, excluding Lyso features) and all BMC plots (Figure S5). Importantly, this revision has not changed our original conclusions comparing the use of Cell Profiler and Harmony in the CPP image analysis process. In addition, per revision #2, we now also provide the corresponding CPP/Cell Profiler profile similarity plots including Lyso features in a new supplementary figure (Fig. S8C). This new Fig. S8C now displays all CPP/Cell Profiler profile similarity plots with/without Lyso features and with/without feature selection side-by-side. We revised the manuscript text and the corresponding figures as follows:

- Update of Fig. 5C to show the CPP/Cell Profiler profile similarity plot (without Lyso features) when feature selection is applied:

Figure 5C. Profile similarity plots showing the correlation of the phenotypic profiles (Spearman correlation of robust z-scores at the feature level, excluding Lyso features) of all reference compounds at each highest non-cytotoxic concentration in MCF-7 cells. Compounds are assigned to one of six clusters (grey-shaded boxes) based on hierarchical clustering. Compounds are color-coded (cyan, orange, red) according to their annotated Actin-, Mito-, or Lyso-related MoA (see Fig. 2A). Colored, dashed boxes highlight correlation scores of compounds with the same annotated MoA. Input feature data (median of all $N_{Tech} = 3$ and $N_{Biol} = 4$ for each feature) are extracted from CPP (CPP/Harmony, CPP/Cell Profiler) or CP (CP/Harmony) images using Harmony or Cell Profiler image analysis software. Correlation scores of negative control compounds (i.e., saccharine, sorbitol) are shown slightly transparent.

- Addition of a new Supplementary Fig. 8C to show all CPP/Cell Profiler profile similarity plots (with/without Lyso features and with/without feature selection):

C Profile similarity plots (robust z-score, feature-level, ex-/including Lyso features, with/without feature selection) of Cell Profiler-based analyses of all reference compounds at highest non-cytotoxic concentrations in different cell lines

Figure S8C. Profile similarity plots showing the correlation of the phenotypic profiles (Spearman correlation of robust z-scores at the feature level, ex-/including Lyso features, with/without feature selection performed) of all reference compounds at each highest non-cytotoxic concentration in MCF-7 cells. Compounds are assigned to one of six clusters (grey-shaded boxes) based on hierarchical clustering. Compounds are color-coded (cyan, orange, red) according to their annotated Actin-, Mito-, or Lyso-related MoA (see Fig. 2A). Colored, dashed boxes highlight correlation scores of compounds with the same annotated MoA. Input feature data (median of all $N_{Tech} = 3$ and $N_{Biol} = 4$ for each feature) are extracted from CPP images using the Cell Profiler image analysis software. Correlation scores of negative control compounds (i.e., saccharine, sorbitol) are shown slightly transparent.

- Addition of a statement referring to the new Fig. S8C from the results/use case 3: *“However, in contrast to CPP/Harmony, those three compounds with annotated Lyso-related MoA (see Fig. 2A) did not form a separate “Lyso cluster” in the profile similarity plots of CPP/Cell Profiler, neither for the reduced set of features nor for the complete set (without feature selection applied) (Supplementary Fig. 8C).”*
- We further added to the discussion: *“Previous reports from several laboratories have also noted the need to perform more extensive filtering of unreliable features when using Cell Profiler^{15,25,27}. From our analysis of CPP data, we concur with this observation, because from the large number of features that were initially extracted using Cell Profiler, ~25% of all features had to be immediately filtered out during the data cleaning procedure due to a high number of missing or identical values. The subsequent feature selection process, which is commonly applied to Cell Profiler*

data,⁴⁹ further led to the additional filtering of many features with high redundancy or low informative value. Although both CPP/Cell Profiler and CPP/Harmony showed the formation of a “Mito cluster”, the three reference compounds with annotated Lyso-related MoA did not form a separate “Lyso cluster” in the profile similarity plots of CPP/Cell Profiler, neither for the reduced set of features nor for the complete set. In that regard, Harmony provided a better separation of the reference compounds. In addition, the required extensive filtering of features made the use of Cell Profiler somewhat less practical overall and may consequently produce different feature lists, which is not optimal when replicate experiments are to be evaluated individually or compared across different screening sites. In turn, when using Harmony, the pre-selected features proved reliable for downstream analysis without the need for additional filtering or feature selection, thereby enabling more consistent comparisons between different screens and laboratories using an identical and complete feature set.”

We are otherwise convinced generally convinced by the arguments of reviewer 2 who argues convincingly that doubling the reagent costs of a primary screen (per the authors own admission) is unlikely to gain wide adoption in a context where the one of the largest benefits of Cell Painting is its low cost and therefore high scalability. We therefore find this manuscript of sufficient technical quality but questionable significant addition to the field.

Answer:

We agree that reagent costs are generally a relevant factor to be considered in HTPP. However, as we already state in the discussion (per 1st revision), it is in our opinion “[...] up to the individual scientist to decide, if the advantages of CPP serve their specific research needs and outweigh a higher investment in cost, time, and resources compared to CP”. In particular, applying CPP to extract a greater phenotypic profile diversity from a single cell can be very relevant for cellular models that are more expensive and laborious than the cancer cell lines commonly used in HTPP. Examples include cellular models that require greater effort for maintenance and differentiation (e.g., iPSC-derived cellular models, neuronal cell lines, etc.) or in cases where availability of raw material is limited (e.g., primary cell lines, patient-derived cells, etc.). Although such more complex cell models, and in

particular 3D models, have not yet been widely used in the conventional HTPP studies, they become increasingly important for phenotypic profiling of larger compound libraries of toxicological and pharmaceutical relevance and for gaining more detailed biological insights.

Moreover, we already state in the manuscript (per 1st revision) that *“The dye concentrations used in CPP and in the original or recently updated CP method were similar indicating comparable screening costs in CPP and CP per single dye used (Supplementary table 1). Thus, the additional reagent costs of the CPP assay are mainly due to the inclusion of the Lyso dye (Supplementary table 1), but may decrease if alternative Lyso dyes compatible with fixed cell staining become available in the future.”*. In fact, we are currently in the process of transferring the CPP method to industrial partners that, e.g., offered to invest money and time into the development of a “ready-to-use” test kit for the CPP dyes and, in this context, also into the development of a cheaper alternative to the Lyso dye that we currently used in this study. We are therefore very optimistic that the reagent cost for CPP will decrease in the future.

Finally, we would like to emphasize that we developed the CPP assay with the intention to enhance the flexibility and customizability of the existing CP-based methods depending on the scientific need and in order to offer additional options for addressing more MoA-specific research questions in the same cell. Moreover, using the Lyso dye is only one option when this organelle is relevant for the study focus. In other cases, it may be replaced by or complemented with a dye labelling, e.g., cellular lipid droplets. Therefore, the concept of performing iterative staining-elution cycles provides the researcher with a higher degree of freedom in terms of dye and target organelle selection, making CPP a versatile approach that we believe does represent a relevant addition to the CP field.

Specifically, to address this issue about the positioning of the CPP assay in the landscape of Cell Painting, we revised the manuscript as follows:

- Discussion: *“In particular, using CPP to extract a greater phenotypic profile diversity from a single cell using iterative staining-elution cycles can be very relevant for research projects using cellular models that are more expensive and laborious than the cancer cell lines commonly used in HTPP. Examples include cellular models that require greater effort for maintenance and differentiation (e.g., iPSC-derived cellular models or neuronal cell lines) or in cases where availability of raw material is limited*

(e.g., primary cell lines, patient-derived cells). Although such more complex cellular models, and in particular 3D models, have not yet been widely used in the conventional HTPP studies, they become increasingly important for generation of human-relevant phenotypic profiling data in both pharmaceutical industry and toxicology.”

Response to reviewer comments on von Coburg et al.

“Cell Painting PLUS: Expanding the multiplexing capacity of Cell Painting-based phenotypic profiling using iterative staining-elution cycles”

Comments received by authors: 24 Feb 2025

Response submitted by authors: 10 Mar 2025

Thank you again for the very constructive feedback from the reviewers. We have revised the manuscript according to their final suggestions.

Reviewer #1:

Specific points:

We thank the authors for adding the figures where CellProfiler features have undergone feature reduction, which allows a more uniform comparison with Harmony features. However, as per previous rounds of review we still disagree with the notion that Harmony features are "more consistent comparisons between different screens and laboratories" since Harmony is an expensive commercial software which only comes on certain brands of microscope, and CellProfiler features can and have been used on Cell Painting data from many brands of microscope, and one can compare data across sites by simply copying another group's final feature list in a single line of Python (or R, etc) code. This is the last though of our technical concerns.

Answer:

We agree with the suggestion of the reviewer. In the revised manuscript, we removed that particular claim:

- **The revised sentence now reads: “In turn, when using Harmony, the pre-selected features proved reliable for downstream analysis without the need for additional filtering or feature selection, ~~thereby enabling more consistent comparisons between different screens and laboratories using an identical and complete feature set.~~”**